



# Relationships between greenhouse gas production and landscape position during short-term permafrost thaw under anaerobic conditions in the Lena Delta

Mélissa Laurent[1], Matthias Fuchs[1], Tanja Herbst[1], Alexandra Runge[1], Susanne Liebner[2,3], Claire C. Treat[1]

[1] Alfred Wegener Institute Helmholtz Centre for Polar and Marine Research, Potsdam, Germany
[2] GFZ German Research Centre for Geosciences, Section Geomicrobiology, Potsdam, Germany
[3] University of Potsdam, Institute for Biochemistry and Biology, Potsdam, Germany

*Correspondence to*: Mélissa Laurent (melissa.laurent@awi.de)

**Abstract.** Soils in the permafrost region have acted as carbon sinks for thousands of years. However, as a result of global warming, permafrost soils are thawing and will potentially release more greenhouse gases (GHGs) such as methane ($CH_4$) and carbon dioxide ($CO_2$). To address the large heterogeneities of GHG releases, this study focused on the relationship between $CO_2$ and $CH_4$ emissions and soil parameters, as well as the evolution of microbial abundance during a permafrost thaw experiment representing the extent of an Arctic summer season. Two depths from three Lena Delta cores taken along a transect from upland to floodplain were incubated anoxically for 68 days at two different temperatures (4°C and 20°C) and an assessment of microbiological abundance ($CH_4$ producers and aerobic $CH_4$ oxidizers) was performed in parallel. Samples from located in upland or slope position remained in a lag phase during the whole incubation, while those from located in the floodplain showed high production of $CH_4$ ($6.5 \times 10^3$ $\mu gCH_4$-C.g$C^{-1}$) and $CO_2$ ($6.9 \times 10^3$ $\mu gCO_2$-C.g$C^{-1}$). Periodic flooding likely allowed the establishment of favorable methanogenic conditions. The presence of higher copy numbers of methanogenic archaea in the active layer of the floodplain than in the upland and slope from the beginning (1.5 to 9.6 times higher) until the end of the incubation time (11 to 700 times higher) supported this hypothesis. In addition, our study pointed out different anaerobic $CO_2$ production (methanogenesis and other respiration) pathways according to landscape position.

**Summary.** Increasing temperatures due to climate change cause permafrost thaw and potentially increasing release of the greenhouse gases $CO_2$ and $CH_4$. In this study we investigated the impact of different parameters (temperature, landscape position, and microbes) on the production of these gases during a short-term permafrost thaw experiment. For very similar carbon and nitrogen contents, our results show a strong heterogeneity in $CH_4$ and $CO_2$ production, as well as in microbial abundance. According to our study, these differences are mainly due to the landscape position and the hydrological conditions established as a result of the topography.



## 1 Introduction

For the past decades, scientists have warned about the effects of global climate change (IPCC 2021). The effects of this warming will be particularly pronounced in the polar regions where the air temperature increase in the past fifty years is already three time higher than the increase in global average during the same period (AMAP, 2021). This particularly affects soils in northern high latitude regions, which contain 1300 Pg of organic carbon (C) (Hugelius et al., 2014). A majority of this C (822 Pg) is stored in permafrost soils (Hugelius et al., 2014), which cover 22% of the Northern Hemisphere (Obu et al., 2019) and

store about 352 Pg of organic C within the first meter. Permafrost is defined as ground where the temperature remains at or below 0 °C for more than two consecutive years (Washburn, 1973). Due to low temperatures the organic matter (OM) stored in permafrost soils is characterized by low degradation rate and permafrost soils exist as a C sink (Hugelius et al., 2014). However, during summer, the upper part of the permafrost thaws (active layer) and allows OM decomposition (Lee et al., 2012).

With climate change, warmer soils and permafrost thaw will likely increase and lead to higher OM decomposition rates due to higher microbial activity. Releases of mineralized C into the atmosphere could reduce the permafrost C pool (Dutta et al., 2006; Schuur et al., 2009) and lead to the transformation of Arctic soils from C sinks to C sources which will further increase climate forcing (Koven et al., 2011; Dean et al., 2018; Lara et al., 2019).

The quality and quantity of OM influence GHG emissions by providing decomposable C (Fox and Cleve, 1983; Hobbie, 2000;

Kuhry et al., 2020). The C is mineralized and released as carbon dioxide ($CO_2$) and methane ($CH_4$) (Wagner et al., 2007; Schuur et al., 2015; Knoblauch et al., 2018).

To quantify $CH_4$ and $CO_2$ emissions and to understand C production from thawing permafrost, numerous incubation studies have been carried out (Lee et al., 2012; Knoblauch et al., 2018; Walz et al., 2018; Holm et al., 2020). They found that $CH_4$ is mainly produced under anoxic conditions; it can also be produced under aerobic conditions, but in much lower quantities

(Schuur et al., 2015; Angle et al., 2017). $CO_2$ is produced under both anaerobic and aerobic conditions. Even though the global warming potential of $CH_4$ is 34 times higher than that of $CO_2$ on a 100 year timescale (Wigley, 1998; Myhre et al., 2013; Neubauer and Megonigal, 2015), under oxic conditions $CO_2$ is released in higher quantity and was considered to contribute more strongly to the permafrost C feedback than $CH_4$ (Schädel et al., 2016). This understanding has, however, changed recently, with one study showing similar production of $CO_2$-C equivalents under anaerobic and aerobic conditions (Knoblauch

et al., 2018). Therefore, C decomposition under anoxic conditions is a major concern. Indeed, warmer temperatures in permafrost-affected soils might lead to wetter soils caused by meltwater from thawing permafrost, and thus to the establishment of anoxic conditions. Nevertheless, it has been shown that not all soils were able to produce the same quantity of $CH_4$ under anoxic conditions, and some were not able to produce $CH_4$ even after several years, e.g., they remained in a lag phase (Treat et al., 2015; Knoblauch et al., 2018). Even though a few factors controlling C decomposition have been identified such as

organic C quantity, temperature, and oxygen availability in soil (Lee et al., 2012; Schädel et al., 2014; Treat et al., 2015; Knoblauch et al., 2018; Ganzert et al., 2007), earlier incubation studies focused mainly on a single temperature and how C





production varies with depth (Lee et al., 2012; Knoblauch et al., 2018; Walz et al., 2018; Holm et al., 2020). Therefore, how different temperatures or landscape positions affect C production under anoxic conditions is not well understood.

Different geochemical and environmental characteristics influence the form and amount of greenhouse gas (GHG) release

from permafrost dominated soils. Temperature (Fox and Cleve, 1983; Neff and Hooper, 2002), wetness conditions, and water table position influence the establishment of anoxic conditions (Morrissey and Livingston, 1992; Whiting and Chanton, 1993). In addition, vegetation stimulates GHG release by providing both a transport pathway from the soil to the atmosphere and a nutrients supply in the form of root exudates, such as glucose, to the microbes which play a key role in the C cycle (King and Reeburgh, 2002a).

Furthermore, the topographic position of field sites was shown to be correlated to C emissions (Treat et al., 2018; Elder et al., 2020). However, as shown by high spatial heterogeneities in C emissions across Arctic landscapes (tundra, wetland, thermokarst, lake) (Virtanen and Ek, 2014; Treat et al., 2018; Elder et al., 2020), it is still uncertain what controls C emissions on a local level (Treat et al., 2018; Lara et al., 2019).  Areas such as drained tundra have the capacity to offset C emissions by acting as C sinks (Juncher Jørgensen et al., 2015; Treat et al., 2018). However, large $CH_4$ emissions have been measured in

low lying wetlands, like floodplains (Bruhwiler et al., 2014; Oblogov et al., 2020). The identification of C hotspots and C sinks throughout Arctic landscapes is necessary to estimate current and future regional C fluxes and to improve our knowledge of the impact of climate change on permafrost-affected soils. However, until now, although such impacts have been identified, they have been little studied in the context of climate change. Previous studies have mainly sought to elucidate the quantity of C emissions released from different landscape forms (Lee et al., 2012; Schädel et al., 2016; Walz et al., 2018), but few studies

have correlated observed heterogeneities in C emissions to landscape position (Treat et al., 2018; Elder et al., 2020).

Besides soil parameters, several studies identified microbial communities as main controls on C emissions instead of the redox conditions established by environmental settings (Liebner and Wagner, 2007; Wagner et al., 2007; Mackelprang et al., 2011) Mackelprang . In particular, methanogenic archaea, which produce $CH_4$, and methanotrophic bacteria, which consume $CH_4$ (Roslev and King, 1996) King, have been detected and identified as crucial for C control in permafrost affected soils (Wagner

et al., 2007; Koch et al., 2009; Knoblauch et al., 2018) Koch, Knoblauch, et Wagner 2009. However, it is still not clear whether microbes or redox conditions exert greater control over C emissions. We started a permafrost soil warming experiment using samples from different landscape positions and incubated the samples at two different temperatures in order to elucidate the effect of different temperatures, landscape positions, and microbial communities on C production.

The aim of this study is to understand and quantify how much C is lost during short-term permafrost thaw across different

landscape units at our model study area in the Lena Delta, Siberia. This study measures GHG emissions based on an incubation experiment and focuses on relationships between GHG emissions and microbial abundance shifts during short-term permafrost thaw under anaerobic conditions. The objectives of the study were to: (1) quantify $CH_4$ and $CO_2$ production during a short-term anaerobic incubation; (2) establish relationships between $CH_4$ and $CO_2$ production and microbes (methanogens and methanotrophs); and (3) identify settings and controls that drive gas production rates in thawed permafrost soils.





## 2 Materials and methods

### 2.1 Site description and sampling

Soil samples were collected in August 2018 on Kurungnakh Island (72.333 N, 126.283 E), Lena Delta, Siberia (Figure 1). Kurungnakh Island is located in the continuous permafrost zone and is an erosional remnant of Late Pleistocene deposits, characterized by ice- and organic-rich sediments (Grigoriev, 1993; Schwamborn et al., 2002) ; most of the island is composed of fluvial sandy sediments and Yedoma Ice Complex (IC) deposits. The IC is made up of ice-saturated sediments (65% to 90%), composed of cryoturbated silty sands and peaty deposits of Holocene origin (Schwamborn et al., 2002; Schirrmeister et al., 2011, 2013). Sediments from the Yedoma IC contain on average 3% total organic carbon (TOC) (Strauss et al., 2013); IC sediments, however, can include organic-rich layers, with TOC content reaching more than 20% in layers with the highest C content.  Thermokarst lakes and wetlands are part of Kurungnakh Island due to thermo-erosional activity (Morgenstern et al., 2021). Samples were also collected in the modern Kurungnakh Island floodplain area. Modern floodplains in the Lena River Delta are of Holocene deltaic origin and are composed of stratified middle to fine sands and silts with layers of autochthonous peat and allochtonous OM (Schwamborn et al., 2002; Boike et al., 2013). The soil sampling was carried out in two stages: first, the active layer was extracted using a spade and active layer samples were collected using a fixed volume cylinder (250 cm$^3$). Then, after excavating the active layer, permafrost soil cores were sampled to a depth of one meter below surface, by drilling with a modified snow ice and permafrost (SPIRE) auger (Jon Holmgren's Machine Shop, Alaska, USA). For this study, three cores were selected due to their location within the local topography: P15, P16, and P17. They were located on an upland, on a slope, and on a floodplain, respectively (Figure 1), with a well-drained upland soil profile. These cores were chosen on the basis of geographical proximity to each other, landscape position, moisture gradient, and ice composition.

Cores were described and subsampled in the field. Detailed core descriptions are presented in Supplementary Table 1. For the purpose of our study, we chose two samples from each core, one from the active layer and another from the frozen layer (Supplementary Table 1). Care was taken not to select samples from the top of the active layer in order to avoid the top organic layers. Cores were subsampled in a climate chamber at -4 °C with a hammer and a chisel instead of a saw, to limit contamination.

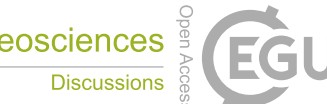

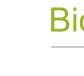

**Figure 1: Location of Kurungnakh Island in the Lena Delta (Siberia). The location of the cores used for the study are indicated on the map and along a schematic transect. Samples were taken during the Lena summer expedition in 2018.**





## 2.2 Sedimentary and geochemical characterization

We characterized the samples for soil texture, C and nitrogen contents, water content, electronic conductivity, and pH. First, samples were thawed at 4° C overnight; then the pore water was extracted with a rhizon soil moisture sampler (Meijboom and van Noordwijk, 1991). Electrical conductivity and pH were measured from pore water for better comparison between samples. Prior to further analyses, soil samples were freeze-dried and the absolute water content (wet weight – dry weight divided by wet weight) was calculated. For TOC, Total Carbon (TC), and Total Nitrogen (TN), subsamples were homogenized and

measured with a carbon-nitrogen-sulfur (CNS) analyzer (Elementar Vario EL III). Each subsample was measured in duplicate, and for each series of measurements, standards and blanks were used to ensure reliable analytical measurements. In order to calculate C and N storage for each sample, bulk density was determined based on a relationship between absolute water content and bulk density (Fuchs et al., 2018) Another subsample was used for grain size characterization. The grain size analysis was conducted using a laser diffraction particle size analyzer (Mastersizer 3000). Prior to measuring, subsamples were put on a

heated shaker for three weeks and $H_2O_2$ was added daily to remove the organics. The samples were measured in a wet dispersion unit and at least three subsamples from each sample were measured. In the end, the average grain size distribution (in vol%) was calculated from the measured replicates.

## 2.3 Incubation set-up and substrate addition

The samples were incubated under anaerobic conditions for 60 days at two different temperatures, 4 °C and 20 °C. For every

sample, three replicates were incubated resulting in a total of 36 samples. Prior to incubation, the samples were thawed at 4°C and prepared under oxygen-free conditions using an anoxic glovebox. The samples were homogenized and 13g of wet soil was collected and inserted into a 120 mL vial. We added sterilized tap water only to samples with a moisture content of less than 30% to limit the effect of gas dissolution (Henry's Law). The flasks were closed with rubber stoppers and aluminium lids. The headspace of the samples was flushed with pure nitrogen for three minutes to remove potential $O_2$ inside the vials. The samples

were incubated in the dark.

After 60 days of incubation, 0.7 mg glucose per gram dry sample weight were added to two of the three replicates to understand the effect of potential substrate limitation in the soil system. The glucose was diluted with milli-Q water to obtain a 100 g.L$^{-1}$ solution. Solutions were injected via syringe to minimize soil disturbance (Pegoraro et al., 2019). The same amount of water as was added with the glucose solution was added to the third replicate to ensure that differences in gas production were only

due to the addition of glucose (Pegoraro et al., 2019; Adamczyk et al., 2021). The glucose addition was also carried out under oxygen-free conditions.

The effects of glucose are usually observed very quickly, which means within less than 48h (Yavitt et al., 1997; Pegoraro et al., 2019). Therefore, after the glucose addition, gas was measured daily for one week. As the first injection had little effect on gas production a second injection (day 64) was added with twice the amount of glucose solution (1.4 mg glucose per gram dry

sample weight).



## 2.4 Gas analyses

$CO_2$ and $CH_4$ in the headspace were measured with a gas chromatograph (GC) (7890A, Agilent Technologies, USA) with flame ionization detection (FID). The temperature in the column was 50 °C with a flow of 15 mL/min and a runtime of 4.5 minutes. Helium was used as a carrier gas. A Hamilton syringe was used to introduce 250 μL of gas into the GC. For the first

week, measurements were made every two days, then twice a week for three weeks, then once a week until day 60. The incubation vials were flushed when either $CH_4$ or $CO_2$ concentration reached $1x\ 10^4$ ppm to avoid gas saturation inside the flask. Finally, the production rate was calculated according to the method of Robertson et al. (1999) and normalized per gram soil C.

The impact of glucose on $CH_4$ and $CO_2$ production was calculated using the cumulative C emissions at 67 days and referred

to a glucose factor.

$$Gf = \frac{(P_{gt} - P_t)}{P_t}$$

Where Gf is glucose factor (%), $P_{gt}$ is total $CH_4$ production rate for samples with glucose, and $P_t$ is total $CH_4$ production rate at i days for samples without glucose.

## 2.5 Quantification of methanotrophic and methanogenic gene copy numbers

Methanotrophic bacteria and methanogenic archaea were quantified with quantitative Polymerase Chain Reaction (qPCR) at different times during the incubations. However, due to laboratory restrictions during the Corona virus pandemic, it has only been possible to analyze 26 samples instead of 90. We decided to analyze only one replicate for each sample at three different times: when the samples were still frozen (1); after 60 days of incubation (2); and after glucose addition (3). For the last point, we selected the two samples with the highest $CH_4$ production rates after the glucose addition among the six samples – the

active layers of P16 and P17. Each sample was replicated three times.

Key genes coding for the enzyme methyl coenzyme-M reductase (*mcrA*) (Thauer, 1998) and for the enzyme particulate methane monooxygenase (*pmoA*) (Theisen and Murrell, 2005) were examined to identify methanogens and methanotrophs, respectively. DNA extractions were performed with a GeneMATRIX Soil DNA purification kit according to the manufacturer's protocol. After DNA extraction, the DNA concentration was quantified by fluorescence with the Qubit dsDNA

HS Assay Kit (Invitrogn, United States). Gene copy numbers were quantified using a SYBRGreen qPCR assay using the KAPA SYBRFAST qPCR Master Mix (Sigma-Aldrich, Germany) on a CFX96 real-time thermal cycler (Bio-Rad Laboratories Inc., United States). All runs were performed in technical triplicates and each run was completed through melt-curve analysis in order to check for specificity of the assay (Liebner et al., 2015). Methanogenic archaea were targeted with the primer set mlas-F/*mcrA*-R (Hales et al., 1996).

To amplify the methanogenic archaea *mcrA* gene, PCR samples were kept at 95 °C for 5 min to denature the DNA. The amplification process was performed with 40 denaturation cycles at 95°C for 1 min, annealing at 60°C for 45 s, and elongating at 72°C for 90 s. To ensure complete amplification, samples were kept at 80°C for 10 min. In addition, to amplify the





methanotrophic *pmoA* gene, using primer pmoA189-F and primer pmoAmb661-R two PCR reaction conditions were used. The first PCR comprised initial denaturation at 95°C for 5 min, 30 cycles with denaturation at 94°C for 45 s, decreasing

annealing temperature from 64°C to 52 °C for 60 s, elongation at 72°C for 90s, and final elongation at 80°C for 90 s. The second PCR comprised an initial denaturation and polymerase activation at 95 °C for 5 min, 22 cycles of denaturation at 94°C for 45 s, annealing at 56°C for 60 s, elongation at 72°C for 90 s, and a final extension at 72°C for 10 min.

## 2.6 Statistical analyses

We used Q10 to measure the temperature sensitivity of the samples. Q10 shows the proportional change in production rates

for an increase of 10 degrees. Q10 was chosen as it is a temperature indicator used in numerous studies (Waldrop et al., 2010; Lupascu et al., 2012; Treat et al., 2015). In addition, for small ranges of temperature such as in our study, Q10 is a reliable indicator (Hamdi et al., 2013). Q10 was calculated via the "equal-time" method, meaning that fluxes from the two temperatures were compared after the same incubation time (Hamdi et al., 2013).

The gas production and microbial data did not show a normal distribution; consequently, it was not possible to test for differences by performing an ANOVA. The differences between cores and depths, and also the impact of temperature on gas production and microbes, were therefore tested using the Kuskal-Wallis test with the R function, kruskal.test().

All statistics and results analyses were performed with R version 4.0.5 (R Core Team, 2021).





## 3 Results

### 3.1 Soil characteristics

All soil samples, except P15-F, were in a pH range of 6.5 -7.5. Most electrical conductivities were very low (<200 µS.cm⁻¹), except for P17-A (635 µS.cm⁻¹; Table 1). Water content was higher in permafrost (54.5%-60.8%) than in the active layer (23.7%-25.8%) for P15 and P16. In contrast, in P17 water content was higher in the active layer (36.2%) than in the permafrost layer (17.2%) (Table 1).

TOC ranged from 0.17% to 3.81%. TOC was slightly lower in the active layer of P16 compared to P15; the opposite was observed for the permafrost layer. Concerning P17, TOC content in the active layer was close to the P15 TOC content. The TOC content in the permafrost layer of P17 was the lowest of the six samples. All the samples had TOC below 6%, and therefore they were considered as mineral soils (%C < 12%) (Table 1) (Soil Survey Staff, 2014).

TN contents were very low for all the samples (>0.3%). TN was below the detection limit of the laser analyzer (below 0.1%) for P17-F. C:N ratios were between 12 and 20. The highest ratios were measured in P15; the lowest were in P17. The C:N ratio was higher in the permafrost layer of P15 than in the active layer.

There were no differences in grain size distribution between P15 and P16. The active layer of P17 contained more clay than the other samples, and P17 was the least sandy sample. In contrast to the active layer, the frozen layer of P17 was the sandiest sample (Table 1).

**Table 1: Chemical and physical properties of the active and frozen layers of the three samples. The conductivity temperature reference was 25°C. Numbers in brackets are standard deviations.**

| Samples | Layer | Landscape position | pH | Conductivity (µS/cm) | Dry Bulk density (g cm⁻³) | Water content (%) | TOC (%) | C/N ratio | Sand (%) | Silt (%) | Clay (%) |
|---------|-------|--------------------|-----|---------------------|--------------------------|-------------------|---------|-----------|----------|----------|----------|
| **P15-A** | Active | Upland | 6.75 | 164.5 | 1.12 | 25.8 | 3.54 | 18.13 | 31.42 | 50.33 | 18.22 |
| **P15-F** | Permafrost | Upland | 6.06 | 150.2 | 0.41 | 60.8 | 2.70 | 20.59 | 28.75 | 53.45 | 17.81 |
| **P16-A** | Active | Slope | 7.21 | 98.6 | 1.18 | 23.7 | 2.70 | 12.95 | 30.09 | 50.8 | 19.13 |
| **P16-F** | Permafrost | Slope | 7.06 | 479 | 0.51 | 54.5 | 3.81 | 12.67 | 26.73 | 55.12 | 18.07 |
| **P17-A** | Active | Floodplain | 7.22 | 635 | 0.88 | 36.2 | 3.48 | 18.46 | 18.89 | 45.40 | 35.72 |
| **P17-F** | Permafrost | Floodplain | 7.44 | 86.4 | 1.36 | 17.2 | 0.17 | | 96.26 | 3.1 | 0.48 |

**Table 2: Summary table of Q10 values, CO₂:CH₄ ratios, and glucose factors. Means of Q10 for CH₄ and CO₂ total emissions after 61 days of incubation. Q10<1 indicates negative effect of temperature on gas production, equal to 1 indicates no effect of temperature on gas production, and Q10>1 indicates positive effect of temperature on gas production. Means of total emission CO₂:CH₄ ratio at 20 °C and 4 °C after 60 days of incubation. Glucose factors were calculated 7 days after glucose addition for each sample with total**
**C emissions. Positive values indicate positive impact of glucose on GHG production and negative values means less CH₄ production after glucose addition.**

| Samples | Layer | Mean Q10 | | Mean CO₂:CH₄ | | Glucose Factor (%) | | | |
|---------|-------|----------|---------|--------------|--------------|-----------|------------|-----------|------------|
| | | CH₄ | CO₂ | 4 °C | 20 °C | CH₄ 4 °C | CH₄ 20 °C | CO₂ 4 °C | CO₂ 20 °C |
| **P15-A** | Active | 0.9 ± 0.5 | 2.4 ± 0.7 | 1455.9 ± 99.9 | 5515.7 ± 2731.9 | -0.10 | -0.38 | 0.02 | 0.18 |
| **P15-F** | Permafrost | 2.6 ± 1.2 | 2.6 ± 1.8 | 1687.7 ± 590.8 | 1544.5 ± 402.1 | 0.02 | -0.31 | -0.20 | -0.44 |
| **P16-A** | Active | 2.7 ± 1.1 | 6.6 ± 3.4 | 2157.6 ± 456.5 | 5168.1 ± 1245.6 | -0.41 | 0.70 | -0.02 | 1.22 |
| **P16-F** | Permafrost | 13.1 ± 22.3 | 6.0 ± 0.9 | 246.1 ± 231.7 | 1710.1 ± 1405.2 | 0.40 | -0.93 | 0.11 | 3.23 |
| **P17-F** | Active | 6006.8 ± 2771.9 | 8.8 ± 3.2 | 707.3 ± 8.1 | 1.1 ± 0.1 | -0.01 | 0.24 | 0.51 | 0.60 |
| **P17-F** | Permafrost | 21.8 ± 10.4 | 3.2 ± 1.6 | 64.2 ± 9.5 | 12.6 ± 11.9 | 1.18 | 0.27 | -0.11 | 0.82 |

## 3.2 Potential gas production

### 3.2.1 Effect of temperature on CH₄ production

Gas production was monitored for 60 days (Figure 2). At the end of incubation, most samples did not show consistent CH₄ production at both 4 °C and 20 °C (Figure 2a; Figure 2b; Figure 3). CH₄ production rates were always below 0.4 µg CH₄-C .g C⁻¹.d⁻¹ and total CH₄ production below 7 µg CH₄-C .gC⁻¹. The active layer of P17 at 20 °C was the only sample that consistently produced CH₄ throughout the incubation. Its lag time ended after 14 days of incubation (Figure 2c) and the maximum CH₄

production rate (355.52 ± 77.16 µg C- CH₄.g C⁻¹.d⁻¹) was reached after 33 days of incubation. Production then stabilized until the end of incubation (Figure 2c). CH₄ production for P17-F-20 started after 47 days of incubation (Figure 2c), for a total amount of 42.53 ± 15.79 µg CH₄-C .g C⁻¹ (Figure 3).

Significant differences in total CH₄ production between cores were only shown at 20 °C (F= Kruskal-Wallis, df = 1, p = 0.0034). CH₄ production in core P17 was higher in the active layer, at both 4 °C and 20 °C, than in permafrost. CH₄ emissions

were larger at 20 °C than at 4 °C for the two depths (F= Kruskal-Wallis, df = 1, p = 0.049). Q₁₀ in P17 for the active layer (6006.76 ± 2771.88) and permafrost layer (21.84 ± 10.38) were consistent with these results (Table 2). P15 and P16 behaved similarly, with higher CH₄ production for the active layer at 4 °C than at 20 °C (F= Kruskal-Wallis, df = 1, p = 0.0065 and F= Kruskal-Wallis, df = 1, p = 0.0374, respectively), and no difference for the permafrost layer. CH₄ production was not found to differ between the active layer and permafrost layer at 20 °C for P15 and P16. However, at 4 °C, CH₄ production from P15

was higher in the active layer than in the permafrost layer (F= Kruskal-Wallis, df = 1, p = 0.04953). Even though the total CH₄ production of P15 and P16 showed differences according to the temperature or to the depth, their CH₄ production rates were very low and therefore, regarding CH₄ production, they were still considered in the lag phase after 60 days of incubation.





### 3.2.2 Effect of temperature on $CO_2$ production

A decrease of $CO_2$ production at the beginning of incubation was observed for all the samples (Figure 2). Overall, temperature

had no impact on $CO_2$ production in the permafrost layers (F= Kruskal-Wallis, df = 1, p = 0.1711) (Table 2, Figure 3b). Concerning the active layers, only P16 and P17 showed a decrease of $CO_2$ production with decreasing temperature (F= Kruskal-Wallis, df = 1, p = 0.0495) (Table 2-Q10). However, after day 33, $CO_2$ production started to decrease for P17-A-20 (Figure 2f). For all cores, $CO_2$ production in the active layer was higher than in the permafrost layer at 4 °C and 20 °C (respectively: F= Kruskal-Wallis, df = 1, p = 0.0152; F= Kruskal-Wallis, df = 1, p = 0.0003) (Figure 3). As with $CH_4$

production, $CO_2$ production of P15 and P16 did not differ under different temperatures. Similarly, the cumulative $CO_2$ release for P17-A, as for $CH_4$, was the highest among all samples (6887.79 ± 933.27 µg $CO_2$-C.g$C^{-1}$) at 20 °C.

The $CO_2$: $CH_4$ production ratio of P17 at 4 °C and 20 °C was in each case the lowest indicating methanogenic conditions. The P17-A-20 $CO_2$: $CH_4$ ratio decreased rapidly during the first 14 days, reached one after 40 days and remained stable until the end of incubation (Table 2). The $CO_2$: $CH_4$ ratio of P17-F-20 at the end of incubation reached 12.55 ± 11.93. At 4 °C the P17-

A $CO_2$: $CH_4$ ratio was 700 times higher than at 20 °C, while for P17-F the ratio was only five times higher (Table 2).





For the other samples, ratios remained high (246.11 ± 231.69 – 5515.66 ± 2731.85) until the end of incubation at both 4 °C and 20 °C (Table 2), which is consistent with the long lag phases and indicates a marginal contribution of $CH_4$-C.

**Figure 2: Gas production at 4 °C and 20 °C for 60 days of incubation. $CH_4$ production of (a.) P15, (b.) P16 and (c.) P17. $CO_2$ production of (d.) P15, (e.) P16 and (f.) P17. Error bars show the deviation from the means ± standard error (n=3). Note differing y-axis scales between cores.**



**Figure 3: Cumulative production of CO₂ and CH₄ after 60 days of incubation at 4°C and 20°C. Scale is expressed as square root in order to have a better display.**

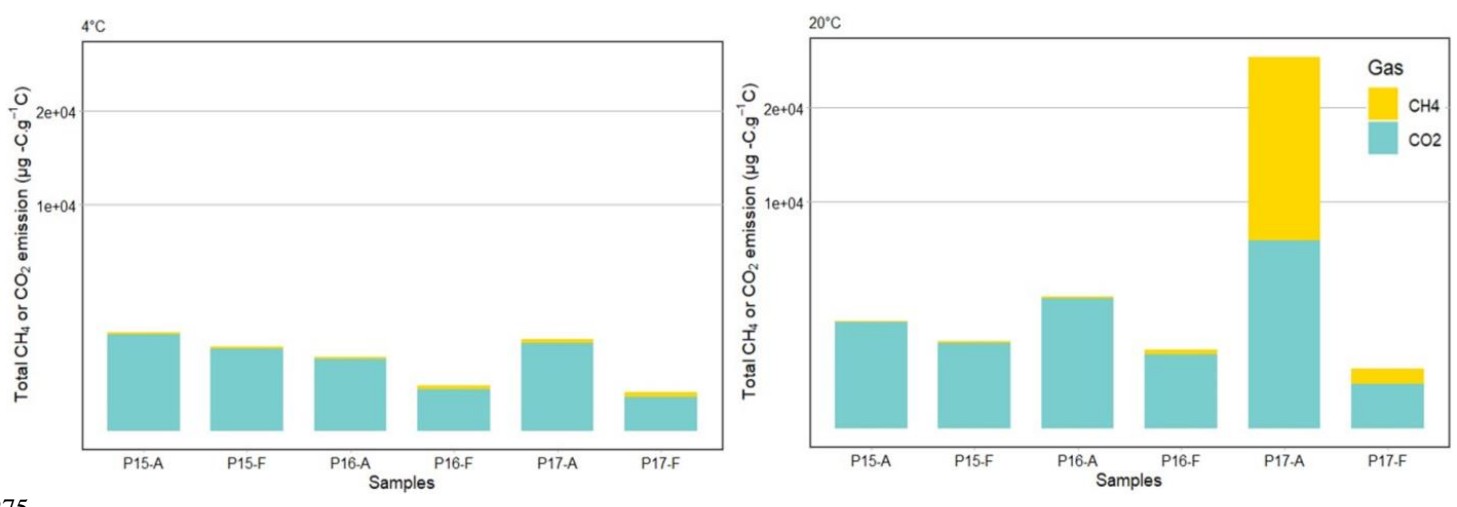


**Table 3: Means of cumulative production of CH₄ and CO₂ (per gram C) at 4 °C and 20 °C after 61 days of incubation (n=3).**

| Samples | Layer | Mean Total CH₄ emissions at 4 •C (µg CH₄-C .gC⁻¹) (n = 3) | Mean Total CO₂ emissions at 4 •C (µg CO₂-C .gC⁻¹) (n = 3) | Mean Total CH₄ emissions at 20 •C (µg CH₄-C .gC⁻¹) (n = 3) | Mean Total CO₂ emissions at 20 •C (µg CO₂-C .gC⁻¹) ( n= 3) |
|---|---|---|---|---|---|
| P15-A | Active | 6.96 ± 1.17 | 1803.48 ± 255.36 | 0.51 ± 0.27 | 2184.38 ± 99.11 |
| P15-F | Permafrost | 1.30 ± 0.35 | 1332.17 ± 494.62 | 0.99 ± 0.32 | 1414.07 ± 141.75 |
| P16-A | Active | 1.61 ± 0.43 | 1012.44 ± 179.93 | 0.66 ± 0.13 | 3309.28 ± 587.99 |
| P16-F | Permafrost | 11.20 ± 8.45 | 340.81 ± 30.54 | 4.34 ± 5.21 | 1074.83 ± 47.79 |
| P17-A | Active | 10.55 ± 1.50 | 1519.87 ± 1052.87 | 6 539.02 ± 1299.21 | 6887.79 ± 933.27 |
| P17-F | Permafrost | 0.49 ± 0.10 | 230.80 ± 8.36 | 42.53 ± 15.79 | 390.60 ± 140.38 |






### 3.2.3 Effect of glucose addition

Glucose factors were calculated together with total $CH_4:CO_2$ production rates (Table 2). Overall, no effect of glucose injection on $CH_4$ production was detected at the end of the incubation period (Table 2) (F= Kruskal-Wallis, df = 1, p = 0.5913). However, a production peak was detected one day after the second glucose addition for P15 and P16. Nevertheless, these variations were very low (0.8 and 9.1%) and appeared only at 20 °C. The variations may be due to the disturbance of the equilibrium due to the dilution of the gas in the water (Henry's law). P16-F-20 without added glucose showed higher $CH_4$ production than with

added glucose. The reason for this is likely the higher $CH_4$ production for this replicate, already observed before glucose addition; therefore, the difference in $CH_4$ production was not correlated to glucose addition. No impact from the glucose addition was detected on $CH_4$ production for the samples at 4°C after either the first or the second injection (Supplementary Figure 2).

While $CO_2$ production at 20 °C was higher for samples that received glucose (Table 2) (F= Kruskal-Wallis, df = 1, p = 0.0192),

no difference in $CO_2$ production was detected for any of the samples at 4°C after glucose addition (Supplementary Figure 2**Erreur ! Source du renvoi introuvable.**). In addition, $CO_2: CH_4$ ratios after 67 days of incubation (with and without glucose addition) were compared (data not shown). No differences were seen between samples with and without glucose addition at both temperatures for all the cores.

### 3.3 Gene copy numbers of methanogens and methanotrophs

We quantified aerobic methanotrophic bacteria and methanogenic archaea using qPCR. Methanogenic gene copy numbers based on the *mcrA* gene ranged from $7.6 \times 10^1$ to $5.85 \times 10^3$ copies per gram wet weight when the samples were still frozen (Figure 4). After 60 days of incubation, the *mcrA* gene copy numbers ranged from $7.62 \times 10^2$ to $5.34 \times 10^5$, depending on temperature. The qPCR results showed significant differences between cores when the samples were still frozen (F= Kruskal-Wallis, df = 1, p = 0.0085) as well as after 60 days of incubation (F= Kruskal-Wallis, df = 1, p = 0.0025). In both cases, P17-

A had the highest copy number per gram soil (Figure 4c). P15 showed no difference between the active and permafrost layer or between 4 °C and 20 °C after 60 days of incubation. P16-F and P17-A had higher copies per gram soil at 20 °C ($4.77 \times 10^4$ and $5.34 \times 10^5$) than at 4 °C ($5.35 \times 10^3$ and $1.49 \times 10^4$) (F= Kruskal-Wallis, df = 1, p = 0.04953). Gene copy numbers after addition of glucose did not differ from those without glucose (Figure 4d).

Gene copy numbers of methanotrophic bacteria based on the *pmoA* gene were mostly between $1 \times 10^3$ and $5 \times 10^3$ copies per

gram soil. No differences were found between P16 and P17 at both 4 °C and 20 °C. Similarly, no difference was found after 60 days. P16-F with glucose had a higher copy number per gram soil than the sample without glucose at both 4°C and 20°C. No difference after the addition of glucose was found for P17-A (Figure 4h). In core P15 *pmoA* could not be detected in any of the treatments, except for the active layer at 4°C. The absence of detectable concentrations is likely due to an insufficient number of microbes in these samples (low DNA concentration) (Supplementary Table 3).








**Figure 4: Means of copies per gram calculated with qPCR amplification at different times, for different conditions - before the incubation (frozen), after 60 days of incubation, and at the end. Gene copy numbers of *mcrA* were calculated for (a.) P15, (b.) (P16), and (c.) P17. *mcrA* results for the active layers of P16 and P17 with or without glucose treatment after 67 days of incubation (d.). Gene copy numbers of *pmoA* are shown for (e.) P15, (f.) P16, and (g.) P17. (h.) *pmoA* results for the active layers of P16 and P17 with**





**or without glucose treatment after 67 days of incubation. Absence of values for some samples is due to either low DNA concentration or failure in qPCR run. Scale is expressed as square root in order to have a better display.**

## 4 Discussion

### 4.1 Overview of different behaviors in GHG production

#### 4.1.1 $CO_2$ production under anaerobic conditions

Anaerobic $CO_2$ production occurred in all the samples, and was similar throughout the cores, except for the frozen layer of P17. $CO_2$ production was slightly higher in the active layer than in the permafrost layer at 20°C only (Table 3, Figure 3). However, overall, none of the variables (landscape position, depth, or temperature) impacted $CO_2$ production.

Nevertheless, $CO_2$ production followed trends in total C and N contents. Indeed, all the samples, except P17-F, had similar C and N contents with values high enough to provide C mineralization (Strauss et al., 2013). In addition, the same samples

produced comparable ranges of $CO_2$ during incubation (Table 1; Supplementary Table 1). Likewise, the frozen layer of P17 showed very low TOC and N contents as well as low $CO_2$ production throughout the incubation.

Under anoxic conditions, $CO_2$ is mainly produced by processes like denitrification or sulfate reduction (Conrad, 1989; Keller and Bridgham, 2007) rather than methanotrophy, which explains why the qPCR results for methanotrophic bacteria and methanogens were very low for P15 and P16 (Figure 4) (Liebner and Wagner, 2007). However, in core P17, the $CO_2$:$CH_4$

production ratio, as well as the presence of high number of methanogenic archaea (Table 2; Figure 4), indicated that $CO_2$ was mainly produced by anaerobic respiration from methanogenesis (Symons and Buswell, 1993; Knoblauch et al., 2018; Holm et al., 2020). Unlike $CH_4$, anaerobic $CO_2$ production can be caused by several diverse anaerobic respiration pathways (Elderfield and Schlesinger, 1998) and mostly depends on C and N content (Knoblauch et al., 2018; Holm et al., 2020), which we observed in our results as well.

In order to simulate effects of root exudates or fresh C, we added glucose. After the addition of glucose, a slight increase of anaerobic $CO_2$ production at 20 °C was shown for P15 and P16. Nevertheless, glucose generally stimulates $CO_2$ production more efficiently under aerobic conditions than under anaerobic conditions (Yavitt et al., 1997; Pegoraro et al., 2019). This may explain the unexpectedly small effect of glucose on $CO_2$ production. Knoblauch et al. (2018) also noticed the small impact of glucose addition on $CO_2$ and $CH_4$ production under anaerobic conditions. In addition, the small effect of glucose treatment on

$CO_2$ production supports the dependence of $CO_2$ production on C and N contents.

High $CO_2$ production rates were shown at the beginning of incubation for all the samples, following by an abrupt decrease. These $CO_2$ peaks are consistent with other studies which have also observed high $CO_2$ production at the beginning of incubation (Lee et al., 2012; Knoblauch et al., 2013; Yang et al., 2021). The rapid C turnover is caused by labile OM immediately available to microbial degradation at the beginning of incubation (Lee et al., 2012; Knoblauch et al., 2013; Yang et al., 2021). High $CO_2$

production in the beginning may also be related to the thawing of samples. Indeed, freeze-thaw activities improve the loss of





soil organic carbon (SOC). The additional SOC is caused by the lysis of dead microbes already present inside the samples (Wang and Bettany, 1993).

### 4.1.2 CH$_4$ production during short term incubation

In contrast to CO$_2$, high CH$_4$ production was detected for only one core, P17, at 20 °C. Under these conditions, the lag time
was substantially reduced and core P17 produced CH$_4$ in both the active layer and the permafrost layer (Figure 3, Figure 3b). In addition, our results indicated a greater CH$_4$ production rate in the active layer than in the permafrost, which is consistent with other studies indicating higher production rates in the active layer than in the permafrost layer (Yavitt et al., 2006; Treat et al., 2015). Unlike P17, both P15 and P16 produced a low quantity of CH$_4$ during incubation at both 4 °C and 20 °C (Figure 3, Figure 3b). Even though total CH$_4$ production was greater at 4 °C than at 20 °C for these cores, CH$_4$ production was still
considered very low (below the blanks; data not shown) and in lag phase.

Knoblauch et al. (2018) also observed long and heterogeneous lag times at 4°C for mineral soils (from 53±23 to up to 2500 days). They explained the long lag time by a lack of methanogens, or a lack of active methanogenic communities in soil samples, which is also applicable to short-term incubations. Lag time is the time required for methanogenic communities to be established in soil. In our study, results from the qPCR analysis supported this theory by showing low methanogen
concentrations over the incubation period and no significant distinctions between 4 °C and 20 °C for P15 and P16 (Figure 4a; Figure 4b). Regarding the active layer of P17, high *mcrA* copies per gram of soil were measured over time (Figure 4), with greater concentration at 20°C than 4°C, which is consistent with Knoblauch et al.'s (2018) conclusions concerning lag time and the observed high CH$_4$ production in this study. Additionally, the high CO$_2$ production in P17 indicated active and abundant microbial communities which corresponds to high copy numbers of methanogens (Table 3, Figure 3; Figure 4c).

Our results showed the absence of a glucose effect on CH$_4$ production rates and on P15 and P16 microbial communities even after the second injection, indicating that the small CH$_4$ production observed was linked to microbe activities rather than to C availability. Regarding P17, we explain the absence of a visible glucose effect by an already high level of CH$_4$ production and overall methanogenic activities. This shows that in mineral soils, glucose is not the factor driving CH$_4$ production.

Overall, this study highlights two different CH$_4$ production behaviors among cores. High rates and quick onset of CH$_4$
production, as well as temperature sensitivity of CH$_4$ production in core P17. Temperature sensitivity is supported by Q10 values (Table 2) and by qPCR analysis (Figure 4), where methanogens were more abundant at 20 °C. Results from Ganzert et al. (2007) also showed that CH$_4$ was produced after one week in floodplain sediments, with greater production rates at higher temperature. In addition, the CO$_2$: CH$_4$ ratio for the P17 active layer at 20°C indicates the establishment of optimum methanogenic conditions by day 40 of this experiment (Symons and Buswell, 1993) (Table 2; Figure 2c). In contrast, P15 and
P16 lagged behind, due to no established methanogenic communities. Even with the addition of glucose both CH$_4$ production and methanogen communities remained below detection limits (Figure 4, Table 2), which supports the lack of active methanogens in these two cores and indicates that the topographic position of the cores is an important factor to consider.



## 4.2 Controls on CH₄ production under anaerobic conditions

The results from soil characteristics showed that the quantity (TOC) and the quality (C:N) of organic C were in the range of
Yedoma deposits and favourable to C mineralization (Zimov et al., 2006) for all the samples (Table 2). Soil characteristics
showed little difference between samples (except for P17-F) (Table 1), hence, they were not able to explain differences in C
production between P15 – P16 and P17 (Figure 3a).

We therefore hypothesized that landscape position rather than soil characteristics played a key role in the establishment of
microbe activities and, consequently, explained variations in GHG production. Regular flooding of P17 and/or a high water
table likely favors the conditions for methanogenesis. Indeed, the methanogen concentration before incubation showed the
highest numbers in the floodplain (Figure 4c). Water saturation allows the establishment of anoxic conditions (Yavitt et al.,
2006) and, therefore, better development of methanogens (Chasar et al., 2000; Paul et al., 2006; Jaatinen et al., 2007; Keller
and Bridgham, 2007). In our case, oxidation marks or redox features were found in the depth profile of core P17, indicating
periodic water saturation under in situ conditions. On the contrary, drier, well-drained conditions in the upland and the slope
inhibit methanogenesis (Megonigal and Schlesinger, 2002). Here we found that a low methanogen concentration existed before
incubation and there was little change in methanogen quantity after 60 days of incubation (Figure 4a, Figure 4b).

When we compare our results to another incubation study (Herbst, 2022) which was carried out using samples from
Kurungnakh Island and nearby Samoylov Island (Figure 1), we found similar results. In the study by Herbst (2022), soil
samples were collected in three different floodplains and incubated for 60 days at 20°C in both aerobic and anaerobic
conditions. Under anaerobic conditions, CH₄ was produced from two of the three cores within 60 days of incubation
(Supplementary Table 2). After 60 days the active layer of the most active floodplain studied by Herbst (2022) ranged around
5 µg C- CH₄.g C⁻¹.d⁻¹, compared to 90 µg C- CH₄.g C⁻¹.d⁻¹ for the active layer of P17. Even if rates from similar floodplains
are lower than what we found, CH₄ production was triggered quickly after the beginning of incubation (from 10 to 40 days).
These results show that floodplain environments allow rapid CH₄ production after permafrost thaw under anaerobic conditions
due to the fast establishment of methanogens. Therefore, the results support rapid establishment of microbes in floodplains
under suitable redox conditions. These results are in line with our hypothesis concerning the impact of landscape position, e.g.
periodically flooded areas provide suitable redox conditions for methanogenesis compared to drier areas. They also support
initially anoxic conditions which trigger CH₄ production, while more aerobic conditions in the landscape coincide with a poor
establishment of methanogenesis even when incubation conditions become favourable for methanogens.

Other in situ studies showed similar high CH₄ production in floodplains compared to drier sites with low CH₄ production
(Huissteden, van et al., 2005; Oblogov et al., 2020). On the one hand, Oblogov et al. (2020) explained high CH₄ production
by wetter conditions due to the floodplain location. On the other hand, Huissteden, van et al. (2005), argued that the high
water table position could be the only wetness condition that could enhance CH₄ fluxes. In our case, no water table was reached
when we cored P17; thus, we conclude that a high water table is not a necessary requirement and periodic flooding can enhance
CH₄ production as well. Huissteden, van et al. (2005) also hypothesized that nutrient supply during flooding could stimulate





methanogens. However, not all floodplains are able to produce high $CH_4$ fluxes (Huissteden, van et al., 2005), and discrepancies between production rates and cumulative emissions of P17 on the one hand, and the data by Herbst (2022) on the other highlight these high heterogeneities regarding $CH_4$ production in floodplains (Supplementary Table 2). Furthermore, settings that lead to high $CH_4$ conditions in floodplains are not fully understood (Huissteden, van et al., 2005) and need further investigations.

In addition to the topographic position, Holm et al. (2020) pointed out that paleoenvironmental conditions strongly drive $CH_4$ production by controlling the establishment of methanogen community. They showed that if paleoenvironmental conditions of soil deposition were favorable to $CH_4$ production, $CH_4$ production during thawing of permafrost, thousands of years later, would be higher than if paleoenvironmental conditions were unfavorable. Our analysis showed a strong dependence on landscape position in the control of C to $CH_4$ mineralization. Moreover, this partly explains the high heterogeneities observed in C production in small areas. Holm et al. (2020) also mentioned the influence of actual environmental conditions on the activity of methanogens. We therefore conclude that even if paleoenvironmental conditions influence the establishment of methanogens after permafrost thaw, suitable actual wetness conditions due to landscape position are an additional control on $CH_4$ production.

## 4.3 Implication for carbon feedback

With climate change, Arctic environments will be subject to changes in moisture conditions, vegetation shifts, and increased active layer depth (Serreze et al., 2000; Hinzman et al., 2005; Myers-Smith et al., 2011). Changes will affect C mineralization differently, depending on landscape position. In our study we identified that $CO_2$ was produced quickly under anaerobic conditions. Treat et al. (2015) studied soils with C and N content similar to our soils, but their soils produced half of the CO2 produced by our samples (except for the permafrost layer of P17). However, in other studies that monitored $CO_2$ production from Yedoma soils, production rates under anaerobic conditions were in a range similar to ours (around 100 µg $CO_2$-C $gC^{-1}.d^{-1}$) (Knoblauch et al., 2018; Walz et al., 2018). These similar results suggest that C in these Yedoma soils is easily available due to the soil's organic-rich characteristics (Strauss et al., 2013). Therefore, our samples exhibited a high $CO_2$ production rate in short-term permafrost thaw experiments, indicating easily available C. However, the same studies, as well as Schädel et al. (2014), pointed out the small size of the labile C pool of Yedoma deposits and nearby soils on Kurungnakh Island (between 5% and 2% TOC content). Short-term incubation studies have relied mainly on the labile C pool (Schädel et al., 2014; Walz et al., 2018; Schädel et al., 2020); therefore, when the labile pool is depleted, carbon production rates likely remain low (Walz et al., 2018).

Our gas production results combined with microbial analysis indicated that $CO_2$ production pathways might change according to the landscape position. In floodplain environments, $CO_2$ production came essentially from methanogenesis (indicated by the 1:1 production ratio, Figure 3) whereas in drier environments, like the P15 and P16 cores, $CO_2$ production also resulted from other, undetermined anaerobic decomposition pathways. Nevertheless, even if pathways were different, $CO_2$ production rates depend mostly on C and N contents (Schädel et al., 2014; Holm et al., 2020). Therefore, landscape position is not a major





factor controlling $CO_2$ production compared to soil characteristics (Figure 5). The $CO_2$ production per gram soil

(Supplementary Figure 1) supports this by showing lower $CO_2$ production for low TOC contents. However, the cumulative $CO_2$ production (per gram C) in the active layer of P17 was two to three time higher than in the other samples, although the other samples had similar C and N contents, with slightly lower N contents in P17 (Table 1). This means that the carbon in the active layer of P17 was more easily decomposed by microbes than the C in the other cores, and that the microbial communities were therefore more active. We suggest that those discrepancies were due to microbe community adaptations under anaerobic

conditions. Even though anaerobic $CO_2$ pathways were established, microbes in P15 and P16 seemed to be less adapted to anaerobic conditions. This could be explained by better drained soils during permafrost thaw in summer for P15 and P16.

Unlike $CO_2$ production, $CH_4$ production is more dependent on landscape position, which triggers methanogenesis. We showed that samples from a floodplain were able to produce a high quantity of $CH_4$ in a short amount of time (less than 40 days) under anoxic conditions at 20°C. Our results and those of Herbst (2022) showed that the active layer of floodplain samples produced

$CH_4$ in large quantity ($6.5 \times 10^3$ µg $CH_4$-C .$gC^{-1}$). Nevertheless, even if $CH_4$ production in permafrost layers started after the production began in the active layers, permafrost layers were still capable of producing $CH_4$. Due to climate change, root exudates, or an additional supply of nutrients from sedimentation of particulate OM (Huissteden, van et al., 2005), will likely increase. In combination with active layer deepening, these landscape locations will maintain or even increase $CH_4$ production in floodplain active layers. Notwithstanding, it is likely that $CH_4$ emissions from floodplains could occur only during flooding

periods. Incubation and in situ measurements have shown that in dry conditions $CH_4$ emissions from floodplain environments were significantly lower than under wet conditions (Huissteden, van et al., 2005; Oblogov et al., 2020). In contrast, even though this incubation experiment was carried out under anoxic conditions, active methanogenic communities were not able to establish themselves in samples from drier areas during a simulated short-term permafrost thaw (Table 3, Figure 3, Figure 4, Figure 5). In addition, in ice-rich permafrost, water saturated conditions are maintained mostly by melt water, whereas

floodplain soils are saturated with water from the rivers which carries nutrients and which likely stimulates microbial activities (King and Reeburgh, 2002b; Oblogov et al., 2020) . Therefore, even if areas like ice-rich tundra reach high water contents due to the thawing of permafrost-affected soils, it is unlikely that a methanotrophic community will be established during a short Arctic summer.

Our experimental study, combined with others, highlights the high potential of $CH_4$ emissions from Arctic floodplains and

allows us to make C feedback predictions of changes in GHG production as a function of landscape position. High $CH_4$ emissions were measured in waterlogged (floodplain) areas while C emissions in uplands mainly came from $CO_2$ production (Huissteden, van et al., 2005; Treat et al., 2018; Oblogov et al., 2020; Hashemi et al., 2021) (Figure 5). Even though we could partly explain heterogeneities of GHG emissions from our incubated samples, it is still uncertain how climate change will impact C emissions under in situ conditions, or in other polar landscapes. Numerous variables and feedback loops such as

vegetation, water table position, flooding time, or nutrient supply to floodplains are still understudied and, therefore, the understanding of C mineralization in permafrost affected floodplains is limited.





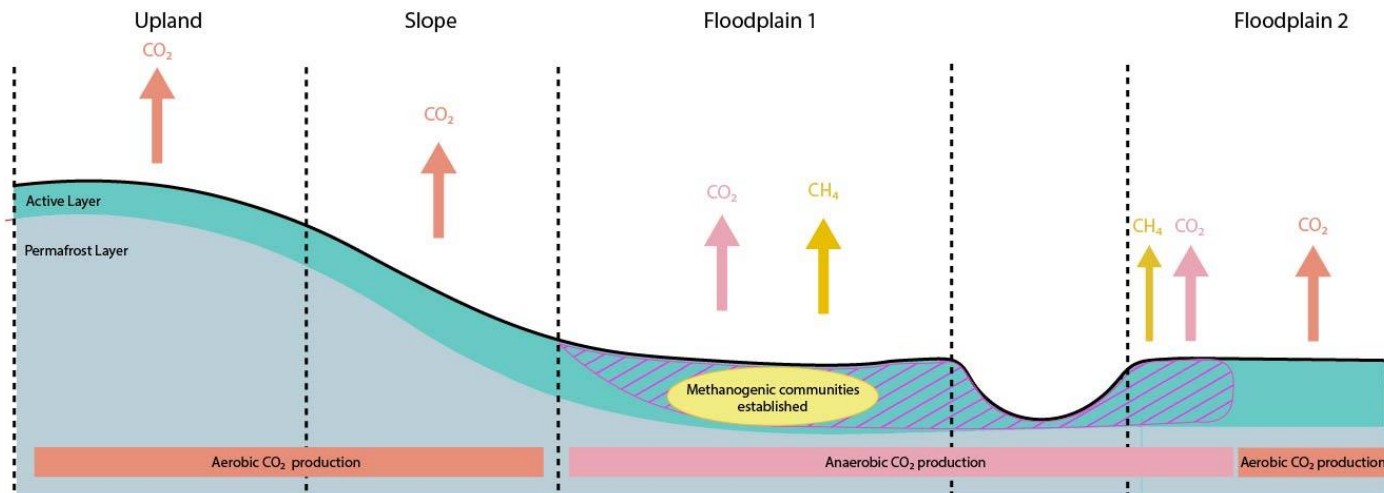

**Figure 5: Schematic figure of the studied transect in warming conditions. Gas emissions are represented according to the results found in this study.**


## 5 Conclusion

In this study, links were made between landscape position, GHG production, and microbes. We observed that C releases as $CO_2$ and $CH_4$ can occur during short-term thawing of permafrost. $CO_2$ was produced in similar quantity from the upland and slope (between 2.2 $9x10^3$ $\mu gCO_2$-$C.gC^{-1}$ and $3.39x10^3$ $\mu gCO_2$-$C.gC^{-1}$), whereas the floodplain produced more than twice that amount ($6.9x10^3$ $\mu gCO_2$-$C.gC^{-1}$). In addition, our study showed that $CH_4$ lag time can be significantly reduced at higher soil temperature if the landscape position favours methanogenesis. Indeed, in a floodplain area $6.5x10^3$ $\mu gCH_4$-$C.gC^{-1}$ was produced, while in the upland and the slope only a slight quantity ($<1$ $\mu gCH_4$-$C.gC^{-1}$) was produced. However, comparisons with other studies showed high heterogeneities for C production in floodplain areas mainly related to wetness conditions (water table position or flooding events). Furthermore, C mineralization in floodplains is largely understudied; therefore, the quantity of in situ C emissions in the context of climate change remains unclear. It is, therefore, necessary to study the different parameters affecting moisture conditions in floodplains (water table position, frequency of flooding) as well as their relationships with microbial species and abundance, and the role of the vegetation in order to achieve a better understanding of processes impacting C losses from floodplains.

### Author contributions

M.L. C.T. and S.L. designed the study. M.L. conducted all the experiments (soil analyses, incubations, and microbe quantification). M.F. and A.R. collected the soil samples and field notes during the expedition in 2018 and created the map. S.L. furnished laboratory materials to perform microbe analyses and gas measurements. T.H. provided data from her incubation experiments. M.L wrote the manuscript with contributions from all the co-authors.

### Acknowledgments

Funding for this study was provided by ERC-H2020 #851181 FluxWIN, the Helmholtz Impulse Initiative and Networking Fund. Samples were collected during the joint Russian-German LENA 2018 expedition to Samoylov Island within the framework of the BMBF KoPf (Kohlenstoff in Permafrost) project (#3F0764B). This project was also supported by the European Erasmus+ programme. We thank the staff at the Samoylov Research Station for support and logistics during the fieldwork. We also thank the Alfred-Wegener Institute and GFZ lab technicians in Potsdam for laboratory assistance.

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
