# Peer review of "Relationships between greenhouse gas production and landscape position during short-term permafrost thaw under anaerobic conditions in the Lena Delta"

_Biogeosciences, 2022_

## Author Comment (AC1)

**Laurent Biogeosciences Reviews & Response to reviewers:**

We thank the three reviewers for their helpful comments and suggestions, which help to improve our manuscript. In response to the thoughtful and constructive comments from the reviewers, we made major revisions to this manuscript throughout all sections and most figures. A summary of these changes is given here, as well as the detailed response to reviewers below. The major changes include:

- 1) Most of the reviewers pointed out that the results would be more relevant and interesting if the incubation time was longer in order to overcome the lag time before methane production that we observed. Unexpectedly, our results showed that after two months of incubation at 20degC, under anaerobic conditions, only one sample layer produced CH4. We originally focused on a short-term incubation because we believe that it is essential to quantify C production under realistic timescale of a growing season (~60 days) in Kurungnakh Island because the aim of this study was to quantify the C production during the growing season under wet conditions communities and identify factors (microbial abundance, substrate availability) that would limit CH4 production in this case study site. However, since only the active layer of the floodplain started producing CH4, we decided to keep the incubation running to see whether the other cores would produce CH4.
- 2) We've included this additional incubation data from days 68 to 363 of the extended experiment. We incorporated additional incubation data into the revised manuscript throughout and have produced a new figure with the cumulative production over a 363-day period. The revised manuscript now shows anaerobic CO2 and CH4 production over a 363-day period.

To summarize the results from this longer incubation period, the floodplain core produced CH4 within the first 60 days due to the already established methanogen communities, as we showed in the initial manuscript. After 6 months of incubation, the permafrost layers from the Yedoma cores started producing CH4. This important result was not included in the earlier manuscript version with the shorter incubation time, as noted by all the Reviewers. Old Figure 4 shows that the permafrost layer in P15 and P16 has a lower methanogen concentration than P17-A, so we attribute the difference in lag time primarily due to the time required for the Yedoma samples to activate the methanogen communities and to produce CH4. We hypothesize that the lack of methanogens in the P16-A and P15-A could be due to the dry condition induced by the landscape position. This indicates that methanogenesis is unlikely established after permafrost thaw in these sediments unless colonized by methanogens and the lack of response of CH4 to the glucose addition and continued anaerobic CO2 production also reduces the likelihood that substrate availability limits CH4 production despite the lower C abundance compared with the floodplain soil (Table 1).

Additionally, the results section has been clarified to distinguish missing versus zero data within the microbial dataset.

- 3) The introduction has been revised to address the concerns of the reviewers, to be more precise and specific about permafrost carbon, the permafrost carbon feedback, and earlier incubation studies. We both narrow the focus to findings from earlier incubation experiments and elaborate on the specifics of the findings regarding the landscape position. In detail, we include a definition, discuss what differs across landscape positions, and expand on the links between microbial abundance and CO2 and CH4 production.
- 4) We substantially revised the methods section to include more details addressing the criticisms of the reviewers and clarified terminology throughout the manuscript (e.g. "production").
- 5) We substantially revised the discussion in order to address the criticism from the reviewers about the overly broad implications of this study despite the limited number of permafrost cores. It is now substantially shorter. We remove Figure 5 (the conceptual diagram) in the revisions. We clarify throughout the manuscript that this is a case study based on permafrost cores from Kurungnakh Island, Siberia, Russia. We shortened and narrowed the discussion to a case study, which aims to understand and quantify the potential C production in this limited region by integrating information about the influence of landscape position, microbial data, and, soil parameters to understand the factors controlling C production in this site within the Yedoma dominated region. We would like to note, however, the importance of these particular permafrost cores collected at a remote field site in Arctic

Siberia, and this data given that it is no longer possible to re-sample at these sites in the foreseeable future.

**Reviewer 1:**

The manuscript of Laurent and co-workers present data from an anaerobic short term incubation study of six samples from three different permafrost affected soils in a transect from ice-complex deposits into a floodplain in the Lena Delta, Russia. The authors incubated the samples at 4°C and 20°C, and measured for 60 days CH4 and CO2 concentrations. At the end of this incubation, they added glucose and measured for another week. Furthermore, they measured the abundance of mcrA genes (methanogenes) and pmoA genes (aerobic methane oxidizers).

We urgently need to better understand the consequences of thawing permafrost in the northern hemisphere on the global carbon cycle. In this respect, the study is concerned with an unquestionable important topic.

However, the main result of the study is that except for one sample, no consistent methane production was observed and that methanogens were still in the lag-phase during the short-term incubation experiment. This means that the experiment was too short to gain information about methanogenesis in most of the samples. Consequently, there is only limited information in the presented Q10 values for methanogenesis or the calculated CO2:CH4 ratios. The remaining results are mainly a confirmation of established knowledge. I suggest that the authors better elaborate, which new information or insights the reader gets from this study.

As explained in the main response, we aimed to simulate wet summer conditions during the growing season. Our results showed that after two months of incubation, only the active layer of the floodplain produced CH4. The absence of CH4 production for the other active layers at 20degC was unexpected. The lack of methanogens and CH4 production after two months show that the methanogen communities were not established. Even though most of our samples did not produce CH4 within the 2 months, we still believe it is important to capture the behaviour of these samples for C production during the growing season. We continued to measure this incubation experiment after the two months presented in this study, as mentioned in the overall summary above. These additional measurements from days 68 to 363 have been added to the study in response to this concern, raised both here and by the other reviewers.

We note that after 6 months of incubation, methane was produced in more samples. Methane was produced consistently across the cores in the permafrost layers at 20degC after 6 months, in both the floodplain (P17) and the Yedoma cores (P15, P16). The active layer of the floodplain core produced methane at both 20degC and 4degC throughout the incubation. However, the active layer of the well-drained Yedoma core (P15) neither produced methane after one year, or at 20degC or with a substrate addition of glucose. The methane production rates of the samples at 4degC were overall very low. The samples that started producing CH4 after one year of the incubation. Therefore, although the methanogenic communities in the Yedoma permafrost layers needed more time to become active, these samples have a high potential for CH4 production after a long thaw period.

This study is a case study in Kurungnakh Island. The geologic history of this site is well known, however few studies have worked on the potential CH4 and CO2 production after permafrost thaw at two temperatures. Here, we contribute to quantify and understand the potential C loss after permafrost thaw along a slope profile from the active and permafrost layers, and compare them with a floodplain in Kurungakh Island. Our findings show that the landscape position plays a key role in the establishment of methanogen communities during the growing season. It is likely that under field condition, only the floodplain produce CH4 during the growing season, this season might not be long enough for the upland and slope to establish methanogen communities. However, with longer thaw time they produce CH4 in the permafrost layer. To our knowledge, these landscape position patterns have not been clearly shown for permafrost sites in Siberia.

Furthermore, the description of methods is in part insufficient to evaluate their suitability

**We have substantially revised the methods based these comments to add further details.**

and the references repeatedly do not support the statement in the text (see detailed comments).

We have substantially revised the methods and the references.

The discussion should substantially be shortened. In its current form its very lengthy, extensively repeats results and itself.

The revised discussion is both shorter and narrower and clarifies the importance of this study as a case study for the Lena Delta region.

The microbial data on methanogenesis are interesting but the importance of the microbial data about aerobic CH4 oxidation remains obscure, since the experiments were done under anoxic conditions.

Thank you for this comment. We agree that the quantification of methanotrophs was not relevant for the study. We removed the data in the revised version.

Finally, I suggest clearly differentiating between production and emission. The data presented here are data on CH4 and CO2 production. There are no data on in situ CH4 and CO2 emissions. Particularly in the discussion, 'emission' is used for both the production in highly artificial laboratory incubations and in situ CH4 and CO2 fluxes. But incubations give only very limited information, if any, about in situ fluxes.

Thanks for this comment. We have changed "emission" by "production" throughout the manuscript unless otherwise necessary.

**Specific comments:**

**L33:** 822 Pg is the C in permafrost, not in permafrost soils. Please clearly differentiate between permafrost (permanently frozen) and permafrost soils (soils containing permafrost).

We revised this sentence and the introduction to specify differences in C stocks between permanently frozen and permafrost soils.

**L34:** Obu et al. 2019 reports that permafrost affected soils cover 14.6% of the northern hemisphere. 21.8% of the northern hemisphere is the permafrost region, i.e. the region where permafrost might be found (but not necessarily underlying 1100% of the soils). Please clarify.

Revised this sentence during revisions and now focus on the extent of the permafrost C stock.

**L38:** Here is a misunderstanding of permafrost. The upper part of permafrost does not thaw in summer, in this case it would not be permafrost (see the definition given in line 34-35).

Revised as suggested: "the upper part of the permafrost affected soils thaws (active layer)".

L44: This sentence is unclear. Who is "providing decomposable C"?

We have significantly revised the introduction. This paragraph is deleted in the revised introduction.

L50: The review of Schuur et al., 2015 does not present data on aerobic CH4 production. Better cite original data.

*Removed during revisions (see major comment #3 above).*

**L79:** The studies cited here report GHG production rates from incubation studies, which do not give much information about 'C emissions released from different landscape forms'. Please clarify.

We have significantly revised the introduction. This paragraph is deleted in the revised introduction (see major comment #3 above).

**L81:** The meaning of this sentence is unclear. Do you mean that microbes with a certain function may be active even if the redox conditions are not suitable for the respective process? Please clarify.

**We have significantly revised the introduction. This paragraph is deleted in the revised introduction.**

**L85:** This is a bit strange question in the context of this study. There are numerous studies on the importance of microbes and redox conditions on e.g. methane production and oxidation, but this study is not addressing redox conditions. Furthermore, in situ C emissions are strongly affected by vegetation, which is not mentioned at all. Please clarify.

**We have significantly revised the introduction. This paragraph is deleted in the revised introduction.**

**L90:** To prevent confusion, I recommend to replace 'emission' by 'production'. In that case, the reader does not expect data on in situ GHG fluxes.

**We have changed emissions to production throughout the manuscript.**

**L133:** Fuchs et al., 2018 determined the bulk density 'by dividing the dry weight of a sample by its original volume'. How may the bulk density be determined by the water content of the soil without knowing the volume of the sample? Particularly when the samples are not water saturated. Please explain.

Thank you for pointing this out. First, we apologize for the wrong reference, which may have caused this confusion. We corrected the reference by "Fuchs 2019". In this thesis, Fuchs plotted the bulk density in relation to the absolute water content of one thousand samples and was able to calculate a transfer function from absolute water content to dry bulk density. To calculate the bulk density, he divided the dry weight of a sample by its initial volume. With the data from the bulk density and the water content, they established a transfer function to determine the bulk density of a sample when the volume is unknown. As we explained in the manuscript, we did not calculate the bulk density, but estimated it according to this transfer function. This transfer function is in the supplementary material XY. In addition, the samples used for our study come from the same area as most of the samples used to establish this correlation. We changed the manuscript and explained the bulk density estimation in more details.

**L162:** Please explain in more detail how the CO2 and CH4 production rates were determined. Did you consider DIC in the soil water? At pH > 7 this might be more than in the headspace. How did you calculate rates from single concentration measurements? I could not find a method in the cited reference (Robertson et al., 1999) that enables the determination of production rates from single gas concentration measurements.

For samples with pH>7, water contents are very low (Table 1), so we assumed that a negligible amount of CO2 was stored as DIC in the sample water. However, we agreed that this might underestimate C mineralization and now mention it in the text.

We did not calculate the production from single gas measurements, but used the change in concentration of CO2 and CH4 over the incubation time. We first converted the concentration from ppm to  $\mu$ g/L using the ideal gas law and then used a linear regression between each measurement point to calculate the change in concentration over time, then calculated the mineralisation rate with the headspace and the volume of the dry content. (after Robertson et al., 1999, Exchangeable lons, pH and Cation Exchange Capacity, p. 266-267)

**L164:** As the equation is written Gf gives the factor by which glucose addition increases gas production, the unit is not %.

The equation has been changed to have %.

**L205:** P16-F has a EC of 479 µS cm-1.

Revised as suggested.

L215: <0.3% L219: ... P17-A .... P17-F

Revised as suggested.

**L301: Could you give the detection limit of your mcrA quantification? Can you measure 76 gene copies per gram?**

The detection limit of the mcrA quantification is 4,3x10+3 copies per gram. We edited Figure 4 by adding the expression "non detected" and "below detection limit".

**L329: Is there a concentration of carbon below which it may not be decomposed? Please explain.**

*C* mineralization depends on the quantity and the availability of the OC. This means that samples with very low *C* content can have high turnover if the *C* is easily decomposable. Therefore, it is complicated to say that there is a threshold below which *C* mineralization does not occur. Here, we clarified that or *C* and *N* contents from our samples are in the same range as those in the study of Strauss et al (2013), and thus, the bioavailability of OC should not be a limited factor for *C* mineralization.

**L332ff:** This is correct as long as sufficient sulphate or nitrate is available, which is generally not the case in terrestrial soils. The reason for low methanogen abundance isprobably rather the high redox potential in these soils.

We deleted this part in the revised version.

**L385: Please explain what you mean by 'favourable to C mineralization'.**

We compared our soil characteristic results with other studies to elucidate whether TOC and N content were limited factors for C mineralization. Other studies showed that with similar, or even lower, TOC and N content C mineralization was possible, and therefore we concluded that C mineralization was not limited by the quality and/or quantity of the OM. Hence, by "favourable to C mineralisation" we mean "not limiting for C mineralisation".

We added more recent studies to the references to support this statement.

**L417:** Which discrepancies do you mean? 'Cumulative emissions' (production) are the consequence of the observed production rates. Please explain.**

We revised this sentence to explain that there is high variability in CH4 production rates within floodplain environments. We compare our results of CH4 production (cumulative production and production rates) with the ones from Herbst (2022). However, the sentence construction is not clear and we have changed it to make it easier to understand.

**L419:** What do you mean by 'methane conditions'. Please explain.

We did not mean "methane conditions", but "methane production". Thanks for noting this mistake.

**L448f**: In a completely anaerobic incubation experiment, landscape position might not be relevant since CO2 production depends on C and N availability. However, at in situ conditions the redox potential differs and hence likely also CO2 production. Please clarify.

**Thank you for this remark. Here, we were referring in anaerobic incubation experiment. We clarified this sentence.**

**Fig 5:** This figure gives no new information or concept. It is quite similar to several figures that have been published previously, even from the same region. Furthermore, the current manuscript gives no information about in situ fluxes. I suggest removing it.

We agreed that this figure was not relevant. We removed it.

**Reviewer 2:**

This paper aims to address some major knowledge gaps on a consequential subject--namely, the controls on potential carbon feedbacks from warming in permafrost regions. These controls are poorly understood due to the many interacting factors (e.g. temperatures, redox conditions, organic matter quality, composition of legacy

microbial communities, etc.) that affect CH4 and CO2 release, and this paper does contribute somewhat to the knowledge base. However, I found this paper to be lacking in terms of the strength of the findings.

First of all, only three cores were analyzed, despite the heterogeneity of the landscape. This is partially compensated by comparisons with other studies, including an extensive discussion of the results in comparison to other cores from the same region analyzed by Herbst (2022).

As explained in the general answer, this study is a case study in Kurungnakh Island. The geologic history of this site is well known, however few studies have worked on the potential CH4 and CO2 production after permafrost thaw at two temperatures. Here, we contribute to quantify and understand the potential C loss after permafrost thaw along a slope profile from the active and permafrost layers, and compare them with a floodplain in Kurungakh Island. Our findings show the landscape position plays a key role in the establishment of methanogen communities during the growing season. It is likely that under field condition, only the floodplain produce CH4 during the growing season, this season might not be long enough for the upland and slope to establish methanogen communities. However, with longer thaw time they produce CH4 in the permafrost layer.

As mentioned above, our study is a case study of Kurungnakh Island. We worked with three cores, but we analysed those cores at two depths and incubated them at two different temperatures. We would like to point out that there is still a lack of data regarding incubation studies in Siberia, and only few studies have been working with the active layer and permafrost layer, and two incubation temperatures. Finally, to establish pan-arctic dataset and compare data throughout the Arctic, smaller studies are essential.

However, the bibliographic entry for Herbst (2022) did not include a link to that manuscript, and I was unable to find it through a web search. Is that manuscript planned to be published in the near future?

The Master's thesis from Herbst (2022) is now available via a permalink on the AWI preprint server EPIC, we added the link in the bibliography section.

Second, the conclusions about microbial abundances do not seem supported by the sparse amount of data shown in Fig. 4. This issue might be partially addressed by better delineating zero abundances vs. truly missing values in the figure, but that depends on how much of the data is actually missing. (see specific comments below)

Thank you for this comment. Based this comment and the one below, we edited Figure 4 and clarified which data was missing and which is zero. We revised the text to clarify that no methanogen was detected for the samples without values and that these should be interpreted as zero values. Methanogen were found only in permafrost layers of P15 and P16, and the active layers of P15-P17. After one year of incubation, the results show CH4 production only for the samples where methanogens were detected (except P15-A).

Finally, the Discussion and Conclusions include numerous statements about how the results can improve predictions of greenhouse gas release under permafrost thaw, but the most significant result (higher methanogenesis in the floodplain active layer) doesn't seem to directly address the effects of permafrost thaw, as the P17-A sample is from the active layer of a floodplain--unless that floodplain location is part of a thermokarst feature, or that sample was formerly part of the permafrost before active layer deepening; but this is unclear from the site description.

Given this comment and the feedback from the other reviewers about the discussion and the shortcomings of the incubation length (major revisions summary #2, #5), we have changed the scope of the discussion to focus more on the relevant results. Additionally, we are now able to quantify the potential effects of permafrost thaw by demonstrating that the lag time before methane production in these thawed upland soils exceeds the length of the current growing season, so CH4 produciton will likely not be an immediate effect of this yedoma thaw under an active layer deepening scenario unless there is some abrupt thaw and landscape change.

**Specific comments:**

li 83-85: Remove extraneous reference text outside the parentheses (3 occurrences).

**Thanks for the comment, we have corrected the text.**

**li 132-133:** After looking at Fuchs et al. (2018), I could not find any information about the "relationship between absolute water content and bulk density."

Thank you for pointing this out. First, we apologize for the wrong reference, which may have caused this confusion. We corrected the reference by "Fuchs 2019". In his thesis, Fuchs determined the bulk density and the water content of one thousand samples from the Lena Delta. To calculate the bulk density, he divided the dry weight of a sample by its initial volume. With the data from the bulk density and the water content, they established a transfer function to determine the bulk density of a sample when the volume is unknown. As we explained in the manuscript, we did not calculate the bulk density, but estimated it according to this transfer function. In addition, the samples used for this study come from the same area as the samples used to establish this correlation. We changed the manuscript and explained the bulk density estimation in more details.

li 142-143: How much sterilized tap water was added to the low-moisture samples?

We calculated the amount of water to add to reach 30% of water according to the water content, and the weight (dry and wet) of the samples. Therefore, the amount of added water differs for each sample.

**li 164-165**: Does the "cumulative emissions" used for calculating the Glucose Factor also include the time before the day 60 glucose additions? If so, this factor might be overly sensitive to random variations in production rates before the additions. (Also, the wording of this sentence is unclear. Suggested rephrasing: "The impact of glucose on CH4 and CO2 production was quantified as a glucose factor, calculated using the cumulative C emissions at 67 days:")

This is a very good point. In the revisions, we will calculate the glucose factor only after glucose addition and see if we have different values. If so, we will modify the results and the discussion based on the results.

**li 167-168:** Related to the above, the phrase "total CH4 production rate at i days" implies an instantaneous rate measured at several (i) timepoints, as opposed to cumulative only at 67 days (from line 164). Which method was used? (If only cumulative, then the phrase "at i days" seems extraneous.)

Thank you for this comment. We used only the cumulative C emissions. Therefore, we deleted the phrase "at i days" and replace the words "production rate" by "production".

**li 173:** I assume "after glucose addition" means at Day 67? Please clarify.

We clarified the text by adding "67 days".

li 202: Typo ("Kuskal-Wallis" should be "Kruskal-Wallis").

Thanks for the comment, we corrected the text.

.li 216: "the lowest [C:N] were in P17": Did you mean P16 (Table 1)?

Yes, that is correct, we meant P16, thanks for your comment.

Table 2: Typo in second-to-last row of first column (first occurrence of "P17-F" seems like it should be "P17-A").

Thanks for the comment, we corrected the text.

**li 247:** It would be clearer to cite Table 3 (from which the  $42.53 \pm 15.79$  value is directly derived) in addition to Figure 3.

Revised as suggested.

**li 251-252**, "P15 and P16 behaved similarly, with higher CH4 production for the active layer at 4 °C than at 20 °C": This doesn't appear to be true for P16, based on its active layer Q10 being >1 (see Table 2).

Thank you for pointing out this mistake. P16-A has higher CH4 production at 20°C. We corrected the manuscript.

**li 253**, "...and no difference for the permafrost layer": This also seems surprising, given that for P16 (Fig. 2b), the blue line (permafrost at 4 degC) is noticably higher than the red (permafrost at 20 degC).

Even though the blue line is higher than the red one, the values of the production rates from the samples at 4°C are still very low and cannot be considered as a real methane production from the samples (samples are still in lag time). In addition, the error bars for the permafrost layer at 4°C and 20°C overlap, meaning that the values cannot be statistically considered as significantly different (supported also with the statistic test). We discussed this in the section 4.1.

Figure 2: Several comments:

- Dashes are missing from the lines in Fig. 2b.

Thank you for this remark. We modified the figure.

- In Fig. 2c, due to the very high Active Layer 20 degC values, it's impossible to see what's happening with the other samples. Would it be possible to create another version of this panel (perhaps for the Supplement) with the very high CH4 values removed, so that the differences in the other lines can be seen?

Thank you for this good remark. We added a zoom in version in the supplementary figures to see the behaviour of the other samples.

- Some of the plots, particularly Fig. 2f, show negative CO2 production rates. How would you explain these?

These negative production rates appear mainly the days where we flushed the samples, therefore it is likely due to the flushing. In the revisions, we replace this figure with one showing the cumulative CH4 and CO2 production where the trends are clearer.

**Figure 4, li 320-321**, "Absence of values for some samples is due to either low DNA concentration or failure in qPCR run.": Can you indicate on the figure (maybe using a symbol) which empty values were due to which cause (low concentration vs. failed qPCR run)? This delineation of zero vs. missing values would help a lot with the interpretation of this figure, as a zero (or below detection limit) concentration still represents the information that concentrations were low, as opposed to not measured at all.

Thank you for this helpful comment. We edited Figure 4 based on your comment, by adding the expression "below detection limit", or "not detected".

**li 330:** Wrong table references for CO2 production (and move "Table 1" reference to line 329 or 331 about C and N contents)?

Revised as suggested, and Figure 2 added to support "low CO2 production throughout the incubation".

**li 390-391**, "methanogen concentration before incubation showed the highest numbers in the floodplain (Figure 4c)": I can't tell whether this statement is supported by Figure 4, as the zero values aren't distinguished from an absence of measurement (see Figure 4 comment above). If all the empty values are actually missing (i.e. due to

failed qPCR), then no direct comparisons of the pre-incubation samples would be possible between P17 and the other sites.

Half of the values from the "pre-incubation" were not measured. In the revised version, we carefully compared the values and specified that the microbial results were in line with the CH4 production, but missing values (likely due to too low concentration) make it difficult to establish a comparison with certainty.

**li 396,** "little change in methanogen quantity after 60 days of incubation": This doesn't appear true for the P16 permafrost layer incubated at 20 degC, which had much higher mcrA (Fig. 4b).

Yes, that is correct. We changed the text according to the comment and specify that this does not apply for P16 at 20degC.

**li 404**, "after permafrost thaw": Are the portions of the floodplains sampled by Herbst (2022) part of thermokarst features?

The samples incubated by Herbst (2022) were not part of thermokarst features. They incubated samples from the floodplains, and samples from the permafrost layer belonging to the same cores.

li 451: Typo; "three time" should be "three times".

**Revised as suggested.**

li 461-463: Invalid sentence structure; did you mean for the end to read "CH4 production will likely increase"?

Yes, you are right, thanks for pointing out this mistake.

li 472, "methanotrophic": Did you mean "methanogenic" ? or both?

*We meant "methanogenic". We changed the manuscript.*

Supplementary Figure 1: Which incubation temperature is shown here (or is it an average of both)?

The incubation temperature shown in this figure is 20°C. We added the temperature to the figure title and the figure.

**Reviewer 3:**

This manuscript presents data from a short-term anaerobic incubation study. The authors present results from six individual samples from three different locations. The goal of the study is to understand the potential effects of temperature, "landscape position", and the addition of glucose to  $CH_4$  and  $CO_2$  production from the soil samples. The premise of the study is interesting and timely.

However, the incubation time of the experiment appears to have been too short, as the methanogens were still in lag-phase. Based on the information presented in the study, I found the links between their results and conclusions unconvincing.

As explained in the main response, we aimed to simulate wet summer conditions during the growing season. Our results showed that after two months of incubation, only the active layer of the floodplain produced CH4. The absence of CH4 production for the other active layers at 20degC was unexpected. The lack of methanogens and CH4 production after two months show that the methanogen communities were not established. Even though most of our samples did not produce CH4 within the 2 months, we still believe it is necessary to capture the behaviour of these samples for C production during the growing season. We continued to measure this incubation experiment after the two months presented in this study, as mentioned in the overall summary above. These

**additional measurements from days 68 to 363 have been added to the study in response to this concern, raised both here and by the other reviewers.**

I think this could be improved be adding more details and specificity to the methods section, in particular.

**We have substantially revised the methods based these comments to add further details.**

More details about the "landscape position" of the sample site would be helpful (i.e. slope, aspect, vegetation cover, etc). I think it would be helpful to consider the scope of the experiment when formulating conclusions.

There is a table with descriptions of the three different sites (see supplementary table 1) in the supplemental materials. In the revisions, we have added pictures of the sites and more details regarding the distance to the rivers surrounding the sample sites. In addition, during the revision, we clarified throughout the manuscript that this is a case study in the Lena River Delta, proposed a hypothesis that these trends may occur at other sites within the permafrost region, and generally narrowed the scope of the discussion and conclusion.

Six samples (from three locations) were incubated for ~70 days. While interesting, there are not enough data points presented in this study to draw meaningful conclusions for permafrost landscapes as a whole.

As mentioned above, our study is a case study of Kurungnakh Island. We worked with three cores, but we analyzed these cores at two depths with laboratory replicates and incubated them at two different temperatures. We would like to point out that there is still a lack of data regarding incubation studies in Siberia, and only a few studies have worked with the active layer and permafrost layer, and two incubation temperatures. Finally, to establish a pan-Arctic dataset and compare data across the Arctic, smaller studies are essential. Nevertheless, we agreed that it is not possible to draw conclusions for permafrost landscapes and use this case study to generate a hypothesis that the landscape position is an important control on the potential for methane production with permafrost thaw in the broader permafrost region.

I strongly suggest that the authors simplify the sentence structure throughout the article. I think that the article can be substantially shortened by removing redundancy and superfluous information/sentences. Most instances of conjunctive adverbs (however, finally, on the other hand, likewise, etc.) should be removed.

Thanks for this feedback. Overall, we found the specific comments from reviewer 3 to be really constructive and helpful particularly in the discussion section. We have carefully revised the manuscript to simplify the sentence structure throughout and to focus it on the question of potential CO2 and CH4 production across different landscape positions in this permafrost landscape.

Also consider the difference between GHG "emissions" and "production" and change your wording accordingly.

We changed "emission" by "production" when necessary and reworked the sentence structure.

Line 10: "release more greenhouse gases". More compared to what? I suggest you remove "more" or be more specific.

**Removed during revisions in order to simplify and be more precise.**

**Line 11:** "to address the large heterogeneities of GHG releases". Spatial heterogeneities? Temporal heterogeneities? I suggest you be more specific here.

**Removed during revisions as above.**

**Line 11:** I suggest you reformulate this sentence. Your study is not really addressing "large heterogeneities of GHG releases". You are trying to understand what the relationship might be between GHG emissions and soil parameters and what factors might be causing large 'spatial' heterogeneities in GHG emissions from permafrost landscapes.

**Removed during revisions in order to simplify and be more precise.**

Line 13: Two depths? Do you mean sediment from two depths from three Lena Delta cores? I suggest you be more specific/clarify.

From each core, samples were collected at two depths. We specified by replacing "two depths" by "Active layer and permafrost layer samples from three cores..."

Line 15: "Samples from located in upland or slope positions". Typos here.

Revised as suggested.

Line 16: Same typo as above "from located in"

Revised as suggested.

Line 18-19: I suggest you rewrite/simplify this sentence and make it easier to read.

**Revised as suggested.**

Line 20: In addition, our study identified different CO2 production...

**Revised as suggested.**

Line 23: Suggestion: Climate change is causing increasing temperatures and permafrost thaw, which might lead to increases in the release of greenhouse gases  $CO_2$  and  $CH_4$ .

**Revised as suggested.**

Line 36: "Due to the low temperatures, the organic matter..."

**Revised as suggested.**

Line 36: The statement that all permafrost soils act as a C sink is misleading. Check out:

Elder et al. 2021 https://doi.org/10.1029/2020GB006922; Anthony et al., 2021 https://iopscience.iop.org/article/10.1088/1748-9326/abc848/meta; etc.

**Removed during revisions.**

Line 40: Consider using an oxford comma throughout the article. It is the standard and will really improve the clarity of your sentences.

**Thank for your comment.**

Line 47: The paragraph beginning on line 47 is two sentences long. Consider merging it with the preceding paragraph.

**Removed during revisions as above.**

Line 53: Consider rewriting this sentence to reduce the number of commas and clauses. Currently, it is difficult to read.

**Removed during revisions as above.**

**Line 57:** Consider eliminating both instances of "able to produce". It is not necessary (i.e., not all soils produced the same quantity of CH4...)

Removed during revisions as above.

**Line 59:** Suggestion: "Even though several factors controlling C decomposition have been ...". I would consider rewriting this sentence to make it more neutral.

Removed during revisions as above.

**Line 61:** What do you mean by "a single temperature". As opposed to temperature profile with depth? Can you be more specific?

Removed during revisions as above.

**Line 62-63:** Suggestion: "Therefore, the relationships between different temperatures, landscape positions, and C production under anoxic conditions are not well understood."

Removed during revisions as above.

Line 62: Consider defining "landscape position". It is not clear to me what you mean by this term. Can you be more specific? Same for "different temperatures". Do you mean the natural spatial heterogeneity of ground temperature in permafrost landscapes (i.e., cooler temps under forest cover, temperature profiles with depth, etc.)?

In the revision, we defined landscape position as a specific geomorphology component of the landscape (e.g. upland, mid-slope, floodplain, as shown in Fig 1b), which affects factors like soil moisture and site drainage. As well, we specified that "temperature" stands for the temperature of permafrost thaw.

Line 64: consider replacing "form" with "type" and "amount" with "quantity"

Removed during revisions as above.

Line 64: released

Removed during revisions as above.

**Line 72:** Consider adding Hughes-Allen et al., 2021 to the list of references as it discusses specifically differences in GHG emissions from different types of thermokarst lakes. https://doi.org/10.1002/lno.11665

Thank you for the reference but we now have removed this paragraph to focus on incubations.

Line 77: I think you are overstating the lack of studies/info here. A quick google search turned up many studies from the last three years describing both experimental studies and in-situ analyses.

Removed during revisions as above.

Line 84: Typo. "King"

Removed during revisions as above

Line 84: Consider finding a different term for "C control". It's not clear what you mean here.

**Removed during revisions as above.**

Line 85: Typo. Citation doubled "Koch, Knoblauch, et Wagner 2009".

**Revised as suggested.**

**Line 86:** You start discussing methods here without yet discussing the objectives of your study. Consider reordering these sentences/paragraphs.

**We revised the introduction to address the objectives of the study first.**

Line 88: You mention landscape positions often, but again, you never define this variable. Please consider defining/being more specific.

Thanks for this critique. We now clearly define this and focus on this aspect in the revised introduction.

Line 89: Define "short term". Days, weeks, months?

Removed during revisions as above.

Line 93: microbesïf microbial community composition? Quantity of microbes? Please be more specific.

Removed during revisions as above.

Line 107: I suggest you make "The soil sampling was carried out..." the beginning of a new paragraph.

**Revised as suggested.**

Line 107: I strongly suggest that you break up this sentence into two shorter sentences. End the first sentence where the colon is.

**Revised as suggested.**

Line 109: You can remove "after excavating the active layer"

**Revised as suggested.**

**Line 112:** Does "with a well-drained upland soil profile" apply to the topography of all three sites? The sentence should be restructured so that it ends with "respectively".

"well-drained upland soil profile", applies for the upland and the slope.

**Line 110-113:** You mention twice that the cores were chosen based on their location within the local topography. I think you can reorder/rework these sentences to make it flow better.

As suggested, we reworked the sentence construction.

Line 115: replace "another" with "one"

**Revised as suggested.**

Figure 1: I suggest that you add Figure sublabels (i.e., a, b) so that you can reference them in the figure caption.

**Thanks for the suggestion. Revised as suggested.**

**Line 127:** "Electrical conductivity and pH were measured from pore water for better comparison between samples." Better comparison compared to what? A different type of method? This sentence isn't super clear to me.

**We revised this sentence in the new version.**

**Line 128:** I think this equation would be more readable if it was presented in normal equation form (i.e., inline equation)

**Revised as suggested.**

**Line 131: how many samples is one series?**

The instruments can both analyse 90 samples for one series. For each analyse (TC and TN), the samples were measured together. We revised as this clearly caused confusion.

**Line 132: Can you describe the relationship?**

Thank you for pointing this out. First, we apologize for the wrong reference, which may have caused this confusion. We corrected the reference by "Fuchs 2019". In his thesis, Fuchs determined the bulk density and the water content of one thousand samples from the Lena Delta. To calculate the bulk density, he divided the dry weight of a sample by its initial volume. With the data from the bulk density and the water content, they established a transfer function to determine the bulk density of a sample when the volume is unknown. As we explained in the manuscript, we did not calculate the bulk density, but estimated it according to this transfer function. In addition, the samples used for this study come from the same area as the samples used to establish this correlation. We changed the manuscript and explained the bulk density estimation in more details.

Line 135: I suggest that you use "organic material" rather than "organics".

**Revised as suggested.**

Line 136: I suggest that you remove "In the end".

**Revised as suggested.**

**Line 142:** I suggest that you keep the passive voice here that you are using throughout the methods. For example, "Sterilized tap water was added to samples with a moisture content of less than 30% to limit the effect of gas dissolution (Henry's Law).

**Revised as suggested.**

Line 152: I suggest you say, "The effects of glucose are usually observed within less than 48h".

**Revised as suggested.**

Line 153: How are you measuring the gas? And do you mean one week as in 7 days or one working week as in days.

**Described in the following section.**

Line 156: Ok now I see the gas section. Maybe just add that it is describe in the following section.

**Revised as suggested.**

Line 162: I suggest that you eliminate "Finally"

Revised as suggested.

Line 172: I suggest that you keep the tone neutral here. Eliminate "We decided". Just explain what you did.

Revised as suggested.

Line 170-175: I think this paragraph can be cleaned up to be more specific and easier to understand.

Based on your comment, we simplified and made this paragraph more understandable.

Line 202: check spelling Kuskal-Wallis

Revised as suggested.

Line 207: I suggest "All soil samples, except P15-F, had a pH between 6.5-7.5."

Revised as suggested.

Line 211: Do you mean TOC weight percent?

Yes, we specified this in the method section.

Line 242: ...CH4 production at either 4°C or 20 °C

Revised as suggested.

Line 242: CH production rates were consistently below...

**Revised as suggested.**

Line 249: I don't think you mean emissions here, rather production

Revised as suggested.

Line 255: Very long sentence. I strongly suggest that you rewrite it to focus on succinctness and clarity.

We reworked the sentence construction to have something more succinct and easier to read.

**Results section:** Limit the results section to the actual results. Currently, you are mixing in some discussion elements. These should really be saved for the discussion section.

**Revised as suggested.**

Line 296: Error in figure cross reference

Thank you for the remark. We corrected the cross reference.

Line 325: I suggest you write 1-2 overview discussion sentences rather than restating the results section.

**Thanks for this feedback, we incorporate this in the newly revised discussion section.**

**Line 325-330:** This section is really heavy on words like "nevertheless, likewise, however, etc." These should be used more sparingly for easier reading. I also believe that you can reduce this paragraph to two sentences.

**Thanks for your comment. We revised the manuscript according to your comment.**

**Line 330-339:** Very nice paragraph and interesting. Can you expand more here, especially the relationship between C and N and anaerobic CO2 production?

*C* mineralization is mainly controlled by the bioavailability of the OM and microbial communities. Under anaerobic conditions, diverse microbial communities are able to decompose the OM to CO2. Therefore, the CO2 production is mainly controlled by the quality (N) and the quantity (TOC) of the OM. Here, our results followed the trend of the TOC contents.

**Line 340: Can you clarify the sentence?**

We expected a large increase of CO2 production rate after the glucose addition. However, only a slight increase of CO2 production was observed at 20degC for P15 and P16. Therefore, we tried to understand why the CO2 production did not increase after the glucose addition.

Line 351: what is lysis?

In this case, lysis means decomposition. We replaced this word in the manuscript as it caused confusion.

Line 352: A concluding sentence would be nice/helpful to wrap up the ideas you present in the preceding section.

Thanks, we revised the discussion to pay careful attention to the sentence and paragraph construction.

Line 363: I think it would be helpful to define lag time much earlier in the paper.

We defined "lag time" after the first use, e.g. section 3.2.1. but because it is important, we discuss this further in the revised introduction.

**Line 373:** I don't think it's appropriate to make this leap from your study to this general statement that glucose availability is not a driving factor for CH4 production in mineral soils.

That is true. We removed this sentence in the revision.

**Line 380-382:** Interesting ideas. Can you expand more here, especially on topographic position? I am not seeing the link between topographic position and the results/factors influencing CH4 CO2 production that you discuss in this section.

In this section we discuss the behaviour of CH4 production under anaerobic conditions. Here, we explain that the lack of CH4 production for P15 and P16 was mainly due to no established methanogen communities. If the methanogen community was small, but established, we would expect to have community growth after the glucose addition. However, since nothing happened, we concluded that this result was an additional support to our hypothesis, e.g., the absence of CH4 production for those samples was because the methanogen was not active (or not active enough). We explain this lack of activity with the actual environmental conditions of sample sites due to the landscape position (cf section 4.2). Based on this feedback, we discuss this point more thoroughly in the revised discussion

Line 385: There are many newer available articles which discuss this subject. Check out

Roy Chowdhury, Taniya & Berns, Erin & Moon, Ji-Won & Gu, Baohua & Liang, Liyuan & Wullschleger, Stan & Graham, David. (2021). Temporal, Spatial, and Temperature Controls on Organic Carbon Mineralization and Methanogenesis in Arctic High-Centered Polygon Soils. Frontiers in Microbiology. 11. 10.3389/fmicb.2020.616518.

**Thank you for the reference, we added more recent references.**

**Line 400-405:** Rather than summarizing the Herbst study so specifically, can you give a more general summary and explain how their result relate to yours and why they might differ?

Yes, that is a good idea. Overall, Herbst et al. (2022) uses samples collected in the same area as ours but exclusively from floodplain soils, and did very similar incubation experiments using anoxic conditions at 20 C. Therefore, we consider this study as a comparison to complete our dataset. The results of Herbst (2022) showed rapid establishment of methanogen communities in floodplains. The lower production rate might be due to lower TOC and TN contents.

Line 406: I suggest "confirm" rather than "are in line with"

**Revised as suggested.**

**Line 410-415:** I think this paragraph can be streamlined and made more concise. Please be specific about how the results/conclusions of the studies you discuss are related to your results.

Thanks for this suggestion. We revised this paragraph to focus on the results from this study, including references to the figures showing these results, the conditions observed in the field. Then we use the observations from other field-based measurements as a comparison. The in-situ measurements show similar trends to what we observed in our incubations. We also compare their explanations for these high CH4 fluxes to our findings.

---

## Referee Report (RR1)

**Line 15:** Yedoma samples located in upland and slope position
I still don't understand what you mean by 'slope position'. Is the sample located on a slope of a certain slope angle (ex. Slope with N aspect and slope angle of 20º?)

**Line 17:** Are the permafrost layer really *producing* permafrost...change wording.

**Line 23:** $CH_4$

**Line 36:** Formatting issue with citation.

**Line 40:** Formatting issue with citation. Check throughout the document, some citations are underlined. Remove "in their study", it is redundant.

**Line 40:** In Treat et al. 2018, they discuss landscape type (upland/wetland), not geomorphology.

**Line 42:** "Landscape position is highly affected by permafrost thaw, low-lying ice-rich permafrost areas can turn out waterlogged environments following permafrost thaw, while higher areas can be drained by water run-off." I don't understand this sentence. Do you mean perhaps that the effects of permafrost thaw are different based on landscape position? Instead of "turn out", maybe you mean "turn into" or "become"?

**Line 51-52:** "Studies have shown that C decomposition depends on several"…

**Line 54:** Treat et al. 2015

**Line 55:** Please define 'landscape categories'.

**Line 56:** Do you mean under anaerobic conditions? ex. Unlike incubation under aerobic conditions, few studies have specifically focused on how … for estimating $CO_2$ and $CH_4$ production under anaerobic conditions.

**Line 57:** Once again, you are using the term "landscape position" frequently in this paper without ever explaining what you mean. Please define this term explicitly and clearly in the beginning of the paper. Do you mean slope angle, slope aspect, elevation, landscape cover? In Treat et al. 2015, they define landscape position as: drained lake basin, active floodplain, wetland, lowland, or upland. Maybe you can use this as an example.

**Line 59:** Consider splitting this sentence into two sentences.

**Lines 63-65:** Please re-read sentences for grammar mistakes and fix.

**Line 70:** Give general geographic precision of Kurungnakh Island.

**Line 96:** Fix grammar (tense) mistake.

**Line 97:** Is a "well-drained upland soil profile" referring to on the upland sample or all three samples? If only the upland sample, this phrase should come directly after the word "upland" or in a separate, clear sentence.

**Line 98:** I don't understand this sentence. Sloping toward which three directions?

**Line 137:** Give exact number of samples which didn't produce $CH_4$.

**Line 226:** Give exact numbers.

**Line 230:** Typo. Change to $CH_4$

**Line 232:** Typo. ºC. This typo appears several times in the rest of the paper. Please double check.

**Line 235:** Over 363 days? So you have continued monitoring $CH_4$ production and you eventually did see $CH_4$ production or are you speculating that one day the samples might start producing $CH_4$?

**Line 251:** Typo 4 ºC.

**Line 256.** Typo

**Line 271:** Statistically significant? If yes give p value, if no, change wording. I believe that the reference is to the incorrect table.

**Line 322:** Do you mean to have several headers under the discussion header? 4.1.1. Different behaviors of what between landscape position.

**Line 361:** What do you mean by "highly constrained"? I believe you mean to say that community size varies highly "between" sites. What do you mean by "narrowness"?

**Line 364:** Typo. 20 ºC.

**Line 373:** Citation missing.

**Line 382:** Delete "On the first hand". Very long sentence with grammar mistakes.

**Line 428:** How much time?

**Line 438:** Dwarf dominated tundra? Dwarf-shrub dominated tundra maybe.

**Line 461:** Typo

**Line 464:** Landscape position is not the trigger of $CH_4$ production.

---

## Author Response (AR2)

Laurent Biogeosciences Reviews & Response to reviewers:

*We thank the two reviewers for their helpful comments and suggestions, which help to improve our manuscript. In response to the thoughtful and constructive comments from the reviewers, we made minor revisions to this manuscript.*

*Notably, we revised the second paragraph in the introduction to explain what we mean by "landscape position"*

Reviewer 1:

I have suggested minor revisions that I hope will improve the clarity and readability of the paper.

Line 15: Yedoma samples located in upland and slope position

I still don't understand what you mean by 'slope position'. Is the sample located on a slope of a

certain slope angle (ex. Slope with N aspect and slope angle of 20º?)

*To clarify, we revised this sentence to read "Yedoma samples from upland and slope cores",*

Line 17: Are the permafrost layer really producing permafrost...change wording.

*We clarified this sentence: "The Yedoma samples from the permafrost layer started producing CH4 after six months of incubation."*

Line 23: CH4

*Revised as suggested.*

Line 36: Formatting issue with citation.

*We checked and solved the formatting issues with citation.*

Line 40: Formatting issue with citation. Check throughout the document, some citations are

underlined. Remove "in their study", it is redundant.

*We checked and solved the formatting issues with citation.*

*Revised as suggested.*

Line 40: In Treat et al. 2018, they discuss landscape type (upland/wetland), not geomorphology.

*We revised this sentence to address this point and to define "landscape position". This now reads:*

*"Treat et al., (2018) showed that C flux variability was strongly associated to specific landscape types due to differences soil moisture and site drainage in uplands and wetlands that also controlled vegetation communities, which we are referring to here as landscape position."*

Line 42: "Landscape position is highly affected by permafrost thaw, low-lying ice-rich permafrost

areas can turn out waterlogged environments following permafrost thaw, while higher areas can be drained by water run-off." I don't understand this sentence. Do you mean perhaps that the effects of permafrost thaw are different based on landscape position? Instead of "turn out", maybe you mean "turn into" or "become"?

*Thanks for pointing this out. We revised the sentence to read:*

*"Landscape change due to permafrost thaw is also highly affected by landscape position: low-lying ice-rich areas can become waterlogged following permafrost thaw, while higher areas can be drained by water run-off (Osterkamp et al., 2009; Liljedahl et al., 2016)."*

Line 51-52: "Studies have shown that C decomposition depends on several"...

*Revised as suggested.*

Line 54: Treat et al. 2015

*Revised as suggested.*

Line 55: Please define 'landscape categories'.

*Kuhry et al., 2021 defined landscape categories by soil type (peaty wetlands, mineral soils) and the origin of the deposits (peat deposit, alluvial deposit). We added this definition in the manuscript. This sentence now reads:*

*"For incubation under aerobic conditions, Kuhry et al., (2020) demonstrated that landscape types based on soil type (peaty wetlands, mineral soils) and the origin of the deposits (peat deposits, alluvial deposits) gave a good estimation of SOM lability, and therefore explained differences in CO2 production better than using only the %C, which is a commonly used metric for C quality across incubation studies (Treat et al., 2015)."*

Line 56: Do you mean under anaerobic conditions? ex. Unlike incubation under aerobic conditions, few studies have specifically focused on how ... for estimating CO2 and CH4 production under anaerobic conditions.

*We split the sentence into two shorter sentences and revised the manuscript as suggested.*

Line 57: Once again, you are using the term "landscape position" frequently in this paper without ever explaining what you mean. Please define this term explicitly and clearly in the beginning of the paper. Do you mean slope angle, slope aspect, elevation, landscape cover? In Treat et al. 2015, they define landscape position as: drained lake basin, active floodplain, wetland, lowland, or upland. Maybe you can use this as an example.

*Thanks for this helpful suggestion. We expanded the explanation of landscape position in the second paragraph (line 42) to read:*

*"Treat et al., (2018) showed that C flux variability was strongly associated to specific landscape types due to differences soil moisture and site drainage in uplands and wetlands that also controlled vegetation communities, which we are referring to here as landscape position."*

*Then we used this helpful suggestion in the 3rd paragraph, the sentence on Treat et al's anaerobic incubation synthesis reads:*

*"Additionally, Treat et al., (2015) highlighted that CH4 production differences were partly explained by the landscape position, with differences seen between uplands, wetlands, floodplain soils, lowlands, and drained lake basins"*

Line 59: Consider splitting this sentence into two sentences.

*Revised as suggested.*

Lines 63-65: Please re-read sentences for grammar mistakes and fix.

*Revised as suggested.*

Line 70: Give general geographic precision of Kurungnakh Island.

*We specified the area (Lena Delta) and the country (Russia) of Kurungnakh Island.*

Line 96: Fix grammar (tense) mistake.

*Revised as suggested.*

Line 97: Is a "well-drained upland soil profile" referring to on the upland sample or all three

samples? If only the upland sample, this phrase should come directly after the word "upland" or in a

separate, clear sentence.

*"well-drained soil profile" refers to the upland only. We changed the wording based on your comment.*

Line 98: I don't understand this sentence. Sloping toward which three directions?

*Deleted in revision as this didn't add much information and was confusing.*

Line 137: Give exact number of samples which didn't produce CH4.

*Eleven of the twelve samples did not produce CH4 for the first two months of incubation. We edited the manuscript based on this comment.*

Line 226: Give exact numbers.

*We specified that four of the six samples produced CH4 at the end of the incubation period.*

Line 230: Typo. Change to CH4

*Revised as suggested.*

Line 232: Typo. ºC. This typo appears several times in the rest of the paper. Please double check.

*Revised as suggested.*

Line 235: Over 363 days? So you have continued monitoring CH4 production and you eventually

did see CH4 production or are you speculating that one day the samples might start producing CH4?

*Here, we refer to the lag time. We continued monitoring CH4 and after 363 days, some samples were still in lag phase, e.g., the CH4 production was still negligible.*

Line 251: Typo 4 ºC.

*Revised as suggested.*

Line 256. Typo

*Revised as suggested.*

Line 271: Statistically significant? If yes give p value, if no, change wording. I believe that the

reference is to the incorrect table.

*We added the p-value and changed the table.*

Line 322: Do you mean to have several headers under the discussion header? 4.1.1. Different

behaviors of what between landscape position.

*We changed the section headers in this discussion section in the revision:*

*4.1. CH4 production in floodplain environment and Yedoma cores across landscape positions*

*4.1.1. Floodplain core*

Line 361: What do you mean by "highly constrained"? I believe you mean to say that community

size varies highly "between" sites. What do you mean by "narrowness"?

*By "highly constrained" we mean that the methanogen communities strongly depend on environmental factors and can not adapt themselves to any environmental conditions.*

*"Narrowness" is used in the same meaning as "highly constrained". Methanogen communities have ecological and phylogenetical functions and it is highly affected by the disturbance of the environment.*

Line 364: Typo. 20 ºC.

*Revised as suggested.*

Line 373: Citation missing.

*We cited Deng et al., 2015 and Ernakovich et al., 2022 two sentences further for this statement. However, based on your comment we added these two citations after "as well as the thaw disturbance".*

Line 382: Delete "On the first hand". Very long sentence with grammar mistakes.

*We restructured the sentence to make it easier to understand.*

Line 428: How much time?

*Schaedel et al 2014 showed that the turnover of labile C pool occurred within the first six months of the incubation. We revised the manuscript as suggested.*

Line 438: Dwarf dominated tundra? Dwarf-shrub dominated tundra maybe.

*Revised as suggested.*

Line 461: Typo

*Revised as suggested.*

Line 464: Landscape position is not the trigger of CH4 production.

*We changed the wording.*

Reviewer 2:

This manuscript has significantly improved since the initial version. In particular, the Discussion and Conclusions are more suitably narrowed to the scope of what was studied, and the presentation of the expanded incubation results (up to 1 year) is interesting. The addition of site photos in Supp. Fig. 1 adds important context to the data presented in the paper. I also like the new Figure 4 much better than the old one—not only due to the better annotation of "not detected" and "below detection limit" data, but also the re-grouping of different field samples under the same incubation treatments, which makes it easier to compare different field sites.

However, there are still a few minor revisions needed before publication, mainly related to correcting typos, but also a few minor content revisions. These are:

*Thanks for the encouraging and helpful reviews. We've adopted the suggestions made here.*

General comment: There is inconsistent use of "," or "." as a decimal separator.

*We carefully changed "," to ".".*

Li 12-14: Sentence structures could use improvement. Suggested revision: "Here, we used an anaerobic incubation experiment to simulate permafrost thaw along a transect from upland Yedoma to floodplain in Kurungnakh Island. Potential CO2 and CH4 production were measured during incubation of active layer and permafrost soils at 4 and 20°C, first for 60 days (approximate length of growing season), and then continuing for one year."

*Revised as suggested.*

Li 18: Change "to trigger" to "triggering"

*Revised as suggested.*

Li 21: Summary is missing opening sentence needed to understand "these gases" in li 22. Copied from tracked changes version: "Climate change is causing increasing temperatures and permafrost thaw, which might lead to increases in the release of greenhouse gases CO2 and CH4."

*Revised as suggested.*

Li 39: Change "differ across Arctic" to "vary across the Arctic"

*Revised as suggested.*

Li 41-43: Revise this sentence to: "Permafrost thaw is highly affected by landscape position: Low-lying ice-rich areas can become waterlogged following permafrost thaw, while higher areas can be drained by water run-off."

*Revised to read:*

*"Landscape change due to permafrost thaw is also highly affected by landscape position: low-lying ice-rich areas can become waterlogged following permafrost thaw, while higher areas can be drained by water run-off (Osterkamp et al., 2009; Liljedahl et al., 2016). "*

Li 62: change "condition" to "conditions"

*Revised as suggested.*

Li 63: change "increase of precipitations" to "increasing precipitation"

*Revised as suggested.*

Li 64: Change "deepen active layer" to "deepening active layer", and "hence" to "and hence".

*Revised as suggested.*

Li ~133-134 (Eq 1 and description): It would make more sense to put this equation above the previous paragraph (i.e. between lines 125 and 126), to be closer to the relevant method text.

*Revised as suggested.*

Li 163: Change "used a linear regression between each measurement point to" to "a linear regression between each measurement point was used to"

*Revised as suggested.*

Li 164-165, "calculated with the headspace and the volume of the dry content and normalized per gram soil C": This wording is unclear. I'm guessing you meant that the headspace concentrations were converted to amounts using the headspace volume and the ideal gas law, and then the gas amounts were normalized to the weight of the dry sample?

*Yes, exactly. We've revised to clarify:*

*"The production rate was calculated with the change in concentration of $CO_2$ and $CH_4$ over the incubation time. First the measured $CO_2$ and $CH_4$ concentrations were converted from ppmv to umol/L using the Ideal Gas Law, then a linear regression between each measurement point was used to calculate the change in concentration over time. The production rate was calculated using the change in concentration over time from the linear regression, then the rates were normalized using the volume of the soil (for differences in the jar headspace) and the weight of the dry soil samples (Robertson et al., 1999). Then these rates were also normalized by the %C found in each sample to look at substrate quality."*

Li 169-170: Going back to my question from the first manuscript version, it sounds like "cumulative" includes the entire time from day 0 to day 67? And what was the result of "calculat[ing] the glucose factor only after glucose addition and see if we have different values" (as mentioned in the Response to Reviews)?

*We compared the two calculation methods and we did not see differences in the values. Therefore, we decided to keep the original calculation method.*

Li 210: Typo in Table 2? It says the TOC of P17-F is 17.2% (old table version said 0.17).

*Thank you for notifying this mistake. We corrected this in revisions.*

Table 2: See comment above about the P17-F TOC. In addition, there is an extra decimal point in this sample's C value, "2..3". Also more generally, maybe it's worth adding TN to the table as its own column separate from C/N, since there's some discussion about it in the text?

*We added the TN values to Table 2 and corrected the TOC value of P17-F.*

Li 242-243: Specify which temperature this applies to, i.e. "With a cumulative 20°C $CH_4$ production reaching…"

*Revised as suggested.*

Li 247-248 "…and the permafrost layer of the same core at 4°C was the lowest": For the P17-F sample, the CO2 production at 20°C actually looks slightly lower than at 4°C.

*You are right. Thank you for your remark. We changed the manuscript based on this comment.*

Li 250-251: Change to "At 4°C, the permafrost layers of the Yedoma core P16 and the floodplain core P17 had cumulative production…"

*Revised as suggested.*

Li 254 & 256: Cite Supp. Fig. 4 here in addition to Fig. 2, since the changes in CO2 production are easier to see there.

*Revised as suggested.*

Li 258-260: These values don't seem to match anything in Supplementary Table 2 (except for the 754 value for P17-F-4).

*Thank you for notifying this mistake. We revised the manuscript with the correct values.*

Li 260-261, "CO2 production plateaued for all the samples": This doesn't appear to be true for P17-A at 4°C (which shows an uptick at the end)?

*You are right. We revised the manuscript with the right values.*

Li 269 and 271: The "Table 2" references need to be corrected to "Table 3."

*Revised as suggested.*

Li 270: The "2.7 ± 2.6 and 2.6 ± 2.1" are slightly different from the values in Table 3.

*Revised as suggested.*

Li 285-286: Change to "CO2:CH4 ratios represent means of total emission after 363 days of incubation."

*Revised as suggested.*

Li 287: Change "less CH4 production" to "less GHG production"

*Revised as suggested.*

Li 296, "0.8 and 9.1": I'm assuming these are absolute amounts (not the glucose factors themselves); what are the units?

Based on your comment, we decided to present the results in different way to make them easier to understand.

*"The response factors were low (CH4 production between 1.2 and 1.7 times higher with glucose addition) and appeared only at 20 °C."*

Li 298, "glucose addition increased CO2 production at 20 °C by 46%": For which sample? Or is this an average?

*Thank you for pointing this out. Tas for the previous comment, we changed the manuscript:*

*"CO2 production at 20 °C was in overall increased by glucose (F= Kruskal-Wallis, df = 1, p < 0.05). The maximum increase of CO2 production was seen for the permafrost layer of P16 (4.2 times higher with glucose addition)."*

Li 302, "core P16-F": Should this be "P17-A" (see upper left of Fig. 4)?

*Revised as suggested.*

Li 318: Delete the extra parentheses around "P16"

*Revised as suggested.*

Li 319: Either delete the "(d.)", or add panel labels to the figure itself.

*We deleted the "(d.)".*

Li 338: Change "did not produce CH4" to "did not produce appreciable CH4" (because Fig. 3 still shows a small amount of CH4).

*Revised as suggested and corrected throughout the manuscript.*

Li 339: Change "4°C and 20°C" to "4°C or 20°C"

*Revised as suggested.*

Li 353: Correct the "Table 2" reference to "Table 3"

*Revised as suggested.*

Li 359: Change "discrepancies" to "variability"

*Revised as suggested.*

Li 363: The word "narrowness" applied to microbial communities needs more clarification. Therefore (and to correct other grammar), change "by the narrowness" to "due to the ecological and phylogenetic narrowness".

*Revised as suggested.*

Li 371: Change "microbial community growth" to "methanogen community growth"

*Revised as suggested.*

Li 373: Change "were correlated" to "is correlated"

*Revised as suggested.*

Li 374: Change to "For ecologically and phylogenetically narrow microbial communities, like methanogens, …" (same reasoning as above; plus other grammar correction)

*Revised as suggested.*

Li 380: Correct missing period at the end of this sentence.

*Revised as suggested.*

Li 382: What are these redox features? Also, change "On the first hand" to "On the one hand".

*Oxidation marks were seen in the field and during the subsampling process. We specified in the manuscript.*

Li 386: Remove the extraneous comma after "both". Also change "that did not produce methane" to "which did not produce appreciable methane" (same reasoning as above).

*Revised as suggested.*

Li 399-405: But the production per gram C would \*always\* be much higher than the production per gram DW, because the C is only a small percentage of the DW in mineral soils. Therefore, comparisons of production per gram C vs. per gram DW doesn't say anything about the lability of the C. To look at C lability, it only makes sense to compare the per-gram-C production across different samples, because most of the rest of the DW is just inert material. This might mean a re-write of this section; e.g. based on Supp. Fig. 3, you could say that the P17-A sample had the highest C lability because it has the highest production per gram C. But P17-F would still have a similar (or slightly lower) lability compared to the Yedoma soils, so its lability is not especially high.

*Thank you for this helpful remark. We re-wrote this section by comparing only the per g C across different samples. It now reads:*

*"Our rates of $CO_2$ production per g C were in the same order of magnitude as other Yedoma incubation studies from Kurungnakh Island (Knoblauch et al., 2013, 2018) and nearby Lena Delta River (Walz et al., 2018). These similar results suggest that C in these Yedoma soils is easily available due to the organic-rich characteristics (Strauss et al., 2013). On the other hand, the adjacent samples from the permafrost layers of the floodplain showed $CO_2$ production g per C similar to the Yedoma cores while, it had the lowest $CO_2$ cumulative production per gram dry weight of soil. Although floodplain environments in the Lena Delta are considered as a low C pool (Siewert et al., 2016), our results showed that the C in the soils was highly labile and comparable to the lability of Yedoma soils."*

Li 409: Change "as proved by" to "consistent with"

*Revised as suggested.*

Li 417: Change "limited" to "limiting"

*Revised as suggested.*

Li 433: Change to "under wet summer conditions, it is likely that there will be rapid C turnover"

*Revised as suggested.*

Li 443: Delete the extraneous "in CH4"

*Revised as suggested.*

Li 448: Change the last part of this line to "soil moisture might increase, and C in Yedoma"

*Revised as suggested.*

Li 451-452: This sentence needs a few small corrections, as follows: "CH4 oxidation in overlying surfaces might have inhibited CH4 production in the active layers of the Yedoma samples (Figure 2; Figure 3), but our methanotroph results did not allow us to draw this conclusion."

*Revised as suggested.*

Li 453-457: But CH4 production in the Yedoma active layers was very low, and occurred only in these anaerobic incubations; whereas in the field the active layers are well-drained and the permafrost is frozen (and therefore not likely actively producing CH4). Therefore, it seems like in the field there wouldn't be much (if any) CH4 *to* transport or oxidize. So I'm puzzled by these assertions about CH4 oxidation and plant transport being important factors in the Yedoma sites, at least while the permafrost is still intact.

*We agree with this remark. Based on it, we discussed more in detail the possibility of not having CH4 production in the Yedoma active layers and removed the parts about oxidation and plant transport that occur in the field but were unfortunately beyond the scope of this experiment.*

Li 465: Change "were" to "was"

*Revised as suggested.*

Li 473: Change to "to better understand changes in redox conditions across the landscape to improve upscaling."

*Revised as suggested.*

Supp. Fig. 5: This figure is hard to read due to the line styles having no particular pattern, and some being very similar in appearance (e.g. the active layer 20C and frozen layer glucose 4C are both solid black). My suggestion would be to use the same set of colors as in Fig. 2 to represent the soil layers and incubation temperatures, and dashed / non-dashed lines to indicate presence or absence of glucose.

*Thank you for this helpful suggestion. We agree that this figure is hard to read. We revised the figure as suggested.*

Supp. Table 2: In the third data column, "Max production rate CH4 (µg C-CO2.g C-1.d-1)", it looks like "CH4" should be "CO2".

*We changed to CH4.*

Supp. Table 3, value "6539.022 ± 1299.21": With these units (mg CH4-C / g C), this would mean that 6.5 times more CH4 was produced than there was C in the soil, which is impossible; and also this number is several orders of magnitude higher than the cumulative production in Supp. Fig. 3c. Maybe a misplaced decimal point? Or are the units for this row supposed to be µg instead of mg?

*Thank you for this remark. We corrected the units to µg instead of mg.*

---

## Author Response (AR3)

Laurent Biogeosciences Reviews & Response to reviewers:

*We thank the editor for the helpful comments and suggestions, which help to improve our manuscript. In response to the thoughtful and constructive comments from the editor, we made minor revisions to this manuscript.*

Lin 95: Which layers? Please specify.

*Based on your comment, we rephrased the sentence:*

*"Sediments from the Yedoma IC contain on average 3% total organic carbon (TOC) (Strauss et al., 2013a)but the TOC content can exceed 20% in organic-rich layers contained within the ice complex sediments (e.g. buried peat horizons, Andreev et al., 2009)."*

Line 125: Provide also details of exact type and supplier.

*We used a 0.6 µm MOM rhizon from Rhizosphere.*

Line 137: Provide also manufacturer, city, [state,] country.

*Mastersizer 3000, Company Malvern, Malvern, UK.*

Line 148/149: Gravimetric or volumetric water content? Please specify.

*We used the gravimetric water content.*

Line 166: Which kind of gas?

*By "gas", we mean the sampling gas. We revised the manuscript based on this comment.*

Line 192: Provide information of supplier, city, [state,] country.

*We revised the manuscript based on your comment:*

*"GeneMATRIX Soil DNA purification kit (Roboklon, Germany)"*

Line 198: Provide information of supplier, city, [state,] country.

*We revised the manuscript based on your comment:*

*"mlas-F/mcrA-R (Microsynth, Balgach, Switzerland)"*

Line 238: What does "this amount" refer to? Please specify.

*We clarified in the manuscript: "The floodplain permafrost core (P17-F) produced 1% of the amount of $CH_4$ produced by the active layer from the same core"*

Line 273: I would expect an F value here, not the name of the test. Delete, if you don't have an F value to report.

*Thank you for this remark. After checking in the literature, we corrected how we reported the results from the Kruskal Wallis test. As recommended, we used the Chi squared value.*

Line 275: Do you mean the value 1? If yes, write "unity" here.

*You are right, we mean the value 1. However, since this value is a ration, it is dimensionless. We edited the manuscript based on your remark:*

*"The P17-A-20 $CO_2$:$CH_4$ ratio decreased rapidly during the first 14 days. The $CO_2$:$CH_4$ ratio reached one after 40 days and remained stable until the end of incubation (Table 2)."*

Line 281: Because you show the square root of the cumulative flux values, you have to indicate that also in the y-axis labels.

Alternatively, you can choose a logarithmic scale, then you don't have to change the y-axis labels.

*We decided to have a square root scale because the cumulative bar plots start at 0. Since log(0) → - ∞ the display of the bar plots do not work with a logarithmic scale. Also, we use the square root scale to have a better display of the data. However, the values showed on the graphs are the real values and not the square root of the cumulative flux.*

Line 294: What does that mean? Please specify.

*Here, we explain that the glucose factors were calculated based on the cumulative C productions, 7 days after the glucose addition.*

*"Glucose factors were calculated based on cumulative C production 7 days after glucose addition."*

Line 327: per gram what? Please specify.

*We revised the manuscript: "Means of gene copies per gram". We changed the y-axis labels.*

Line 331: As above, why not using a logarithmic scale?

*As for comment Line 281, we use cumulative bar plots starting at 0 and the values showed on the graph are the real values.*

Line 350: What do you mean with difficult? Please specify.

*By "difficult", we mean, "non suitable conditions". We added a citation to support this statement: (Eskelinen et al., 2009)*

Line 368: Between what? Please specify.

*Here we, compared lag times and CH4 production rates from several previous studies. The lag times and CH4 production highly differed between the studies.*

"As explained above, lag times measured from former studies differed".

Line 384: What is meant here with disturbance? Length of thawing period? Please specify.

*Yes, by thaw disturbance we mean the length of thawing period (brutal thaw or sequential thaw).*

Line 389: What does that mean, specify.

*Here, we explain that since methanogen communities are ecologically and phylogenetically narrow microbial communities; the establishment of their community is highly controlled by random environmental factors (stochastic processes).*

*We defined this term in the manuscript line 385: "For ecologically and phylogenetically narrow microbial communities, like methanogens, random environmental processes like microtopography (stochastic processes)"*

Line 413: What do you mean here? A pool with low C content? Or a small C pool? Please specify.

*Here, we mean low C content pool. We revised the manuscript based on this comment.*